# Base Point Split Algorithm for Generating Polygon Skeleton Lines on the Example of Lakes

**Elżbieta Lewandowicz ***  **and Paweł Flisek**

Department of Geoinformation and Cartography, Institute of Geodesy and Civil Engineering, Faculty of Geoengineering, University Warmia and Mazury in Olsztyn, 10-719 Olsztyn, Poland; pawel.flisek@student.uwm.edu.pl
* Correspondence: leela@uwm.edu.pl; Tel.: +48-895234467

**Abstract:** This article presents the Base Point Split (BPSplit) algorithm to generate a complex polygon skeleton based on sets of vector data describing lakes and rivers. A key feature of the BPSplit algorithm is that it is dependent on base points representing the source or mouth of a river or a stream. The input values of base points determine the shape of the resulting skeleton of complex polygons. Various skeletons can be generated with the use of different base points. Base points are applied to divide complex polygon boundaries into segments. Segmentation supports the selection of triangulated irregular network (TIN) edges inside complex polygons. The midpoints of the selected TIN edges constitute a basis for generating a skeleton. The algorithm handles complex polygons with numerous holes, and it accounts for all holes. This article proposes a method for modifying a complex skeleton with numerous holes. In the discussed approach, skeleton edges that do not meet the preset criteria (e.g., that the skeleton is to be located between holes in the center of the polygon) are automatically removed. An algorithm for smoothing zigzag lines was proposed.

**Keywords:** straight skeleton; hydrographic network; network modeling

## 1. Introduction

Automated solutions for processing vector data relating to surface water bodies and rivers have many practical applications, and they can be used to generate a skeleton of a hydrographic network. This is accomplished with the use of generation algorithms. Various generation methods have been described in the literature, including the following:

- Medial axis transformation [1–3];
- Chordal axis transform [4];
- Straight skeleton [5,6];
- Delaunay triangulation [6–9];
- Splitarea algorithm [10,11];
- Other [12–14].

In the described methods, the process of plotting skeleton lines begins with the generation of a base skeleton. In successive steps, the selected skeleton edges are modified and removed to improve the applicability of the results. Skeletons are modified with the use of various methods, including correction [7,15], extraction [8], hierarchical feature extraction [16], filtration [17,18] and generalization [19]. Different types of triangles are identified in polygons that are generated with the use of methods based on triangulated irregular networks (TIN) [7,10,20]. Different approaches to generating skeleton edges for various triangle types have been proposed in the literature [7,10,20]

(Figure 1). In a type 1 triangle, one side is a section of the polygon boundary line, and the two remaining sides intersect the polygon. In a type 2 triangle, two sides are segments of the polygon boundary line. The vertices of a type 3 triangle are located on the polygon boundary line, and none of the triangle's sides overlap the polygon boundary. The above also applies to a type 4 triangle.

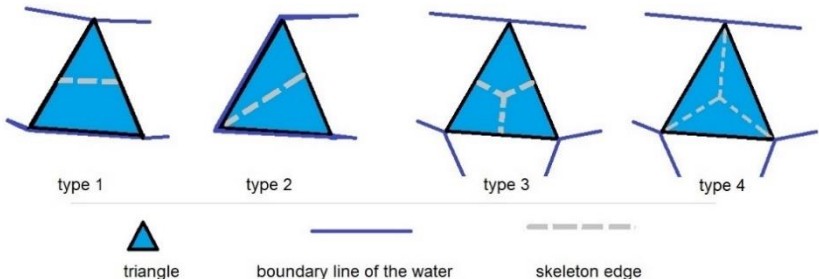

**Figure 1.** Triangle types: type 1—link triangle, type 2—ear triangle, types 3 and 4—branch triangles.

In the literature [8,10,14], skeletons were generated based on an analysis of a polygon's surroundings, and neighbors were identified on the right and left side of polygon boundaries. Neighbors are not always easy to identify, and in some cases, the analyzed area has to be expanded [10]. This process is particularly difficult when complex polygons contain holes (islands on water bodies).

In a previous study [14], the authors applied a simplified method to generate the centerline of an elongated polygon on the example of a river island. Topological data describing the neighborhood of the polygon representing a river channel were used to select TIN edges intersecting the river channel in a direction perpendicular to river flow. The presence of an island on the river complicated the procedure because the neighborhood relations between the island boundary and the left and right boundary of the river channel had to be considered. The authors concluded that the proposed algorithm should be tested and modified in analyses of other objects with varied shapes, in particular in networks composed of a large number of connected water bodies [14] (p. 17).

Such attempts have been made in the current study to eliminate the shortcomings of the previously described algorithm, where the neighborhood relations between polygon boundaries and the neighboring polygons had to be taken into account. This task proved to be particularly challenging in polygons containing numerous holes.

The aim of this study was to develop an algorithm for generating skeletons of complex polygons that represent elements of a complex hydrographic network, such as a system of lakes connected by rivers or lakes with numerous islands. Other flow lines, such as rivers flowing through lakes, are also identified in the hydrographic network. For such lines to be represented in the polygon skeleton, the points on the lake boundary marking the river inflow and outflow should be taken into consideration in the algorithm. In the proposed solution, these points were referred to as base points (BP). Base points significantly influence the shape of the polygon skeleton. In the discussed solution, an unlimited number of base points can be identified on the boundary of a complex polygon representing a lake. The only requirement is that base points are located on the polygon's boundary line because they determine the shape of the generated skeleton. The importance of base points in the algorithm was reflected in the title of this paper.

The developed algorithm was applied in several research tasks. The first task was to investigate whether the designed algorithm can generate an appropriately shaped skeleton at the beginning of the process, and whether it can be applied in various cases. The shape of the resulting skeleton was determined by the location of base points on the boundary of a complex polygon (boundary of a water body). These points are important for the practical application of the generated skeleton. The identified points will be the hanging nodes in skeletons of complex polygons. The skeleton will be used to create a geometric model of a hydrographic network for the purpose of hydrographic modeling or navigation. The second research task was to investigate the ability of the proposed algorithm to generate skeletons

for polygons with numerous holes (islands on water bodies), where the skeleton's location between islands was modified automatically based on the preset criteria. The third task involved the search for a solution where the skeleton could be generalized by eliminating zigzag effects. The modified moving average approach was used.

The present study was conducted with the use of programming tools in ArcGIS (ESRI) software [21,22] and the algorithms developed by the authors in Python.

## 2. Methods and Materials

### 2.1. The First Research Task—Characterization of the Bpsplit Algorithm

The first research task relied on the assumption that the expected shape of a skeleton could be defined at the stage of data preparation. The analyzed polygon represents a water body, and it is one of the elements of a hydrographic network composed of lakes connected by a river. The hydrographic network was developed based on vector data. The Base Point Split (BPSplit) algorithm was created. Base points that significantly influence the generated skeleton were defined in the first step. The edges of the Delaunay triangulation (triangulated irregular network, TIN) model inside the polygon were the key elements of the algorithm. In classical solutions involving TIN models, the skeleton is generated based on all TIN edges inside the polygon with the use of solutions that are applied to various types of TIN triangles (Figure 1). The skeleton is then modified [10].

In the authors' previous study [14] and in the current study, the selected TIN edges were used to generate a skeleton. In the previous study, TIN edges were selected based on the topological relations between polygon boundaries and the neighboring polygons. In this study, TIN edges were selected based on the adopted set of base points. Base points were localized on the boundary of a polygon, for example, at the source or mouth of a river intersecting a lake. Base points constitute hanging nodes in the resulting skeleton.

In the developed algorithm, base points divide the polygon's boundary line into segments. Each boundary segment begins and ends in a base point. The segmentation technique is used to select TIN edges. TIN edges that touch different segments of the boundary line are selected from the set of TIN edges located inside the polygon. TIN edges that touch base points are removed from this subset. The midpoints of the selected TIN edges and base points constitute a basis for generating the skeleton of a hydrographic network. Additional data are required to combine these points into a skeleton. In classical TIN methods, a skeleton is generated with the use of the structures defined for different types of triangles (Figure 1). Triangles are not analyzed in the proposed solution. The selected TIN edges are used to divide the polygon (for example, a polygon representing a lake) into a set of smaller polygons. Skeleton edges are created based on the midpoints of the selected TIN edges and base points. The edges are generated between the points located on the boundary of a single polygon segment.

The proposed algorithm is an improved version of the previously proposed solution [14]. The developed algorithm is presented below in graphical form in Figure 2. The input data were the complex polygon and the set of base points. The set of base points is prepared by the user.

The method of selecting TIN edges (7) that touch two different segments of the boundary is an important consideration in the BPSplit algorithm. A simple solution was proposed to fulfill this requirement. In the previous solution [14], TIN edges were selected by identifying the topological relations between boundary lines and the adjacent areas. This solution is laborious in lakes with a large number of islands. In the proposed algorithm, TIN edges are selected based on simple topological relations. The relations between the boundary lines of a complex polygon and the neighboring areas do not have to be determined.

The two proposed tools that were applied in the algorithm are described below.

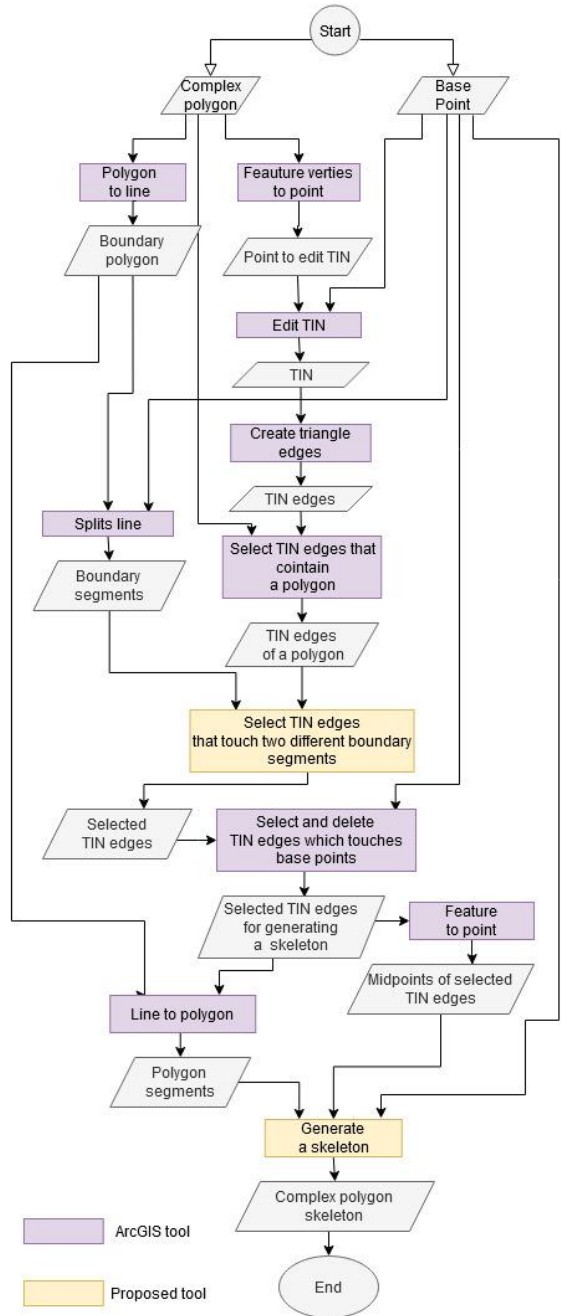

**Figure 2.** Flowchart of the Base Point Split (BPSplit) algorithm.

2.1.1. Method of Selecting TIN Edges which Touch Two Different Boundary Segments in the BPSplit Algorithm

The process of selecting TIN edges plays an important role in the BPSplit algorithm. TIN edges are selected based on segments of polygon boundaries. TIN edges that touch different boundary segments are selected. The relations between TIN edges and polygon boundary segments have to be determined for this purpose. This is accomplished by analyzing topological relations during data processing. A set of points that constitute the vertices of TIN edges is generated. These points are assigned the identifiers of TIN edges (TIN_ID) in attribute tables (Figure 3b).

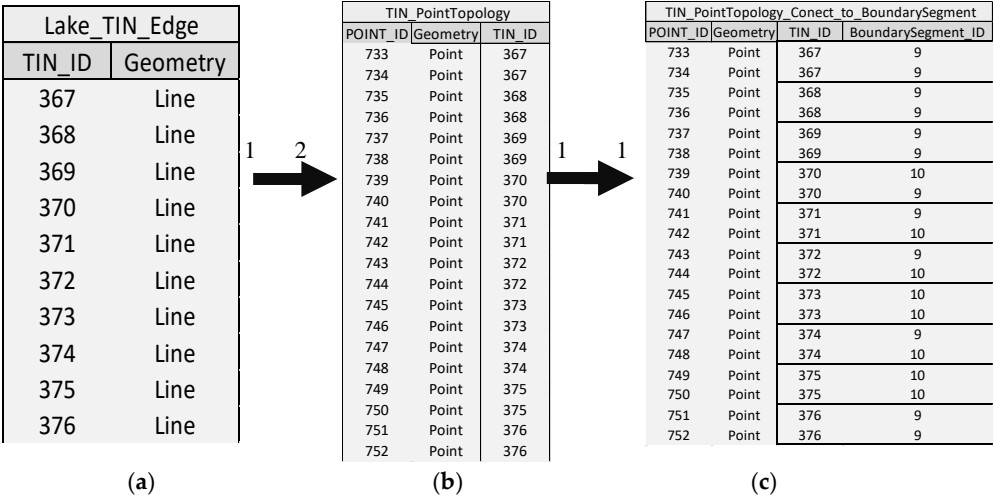

**Figure 3.** Topological relations during data processing: (**a**) attribute table of triangulated irregular network (TIN) edges; (**b**) attribute table of TIN edge vertices with Vertex ID (Point_ID) and TIN_ID values; (**c**) attribute table of TIN edge vertices that touch boundary segments (containing the values of BoundarySegment_ID).

The resulting data set contains twice as many vertices as TIN edges. As a result, the attribute table contains two points with identical TIN_ID (Figure 3b). In the next step, the algorithm connects boundary segments with TIN vertices that touch them. The ID of the boundary segment (BoundarySegment_ID) that touches TIN vertices is included in the attribute table (Figure 3c). These data are used to select the ID of TIN edges touching different boundary segments. The algorithm compares the values of TIN_ID and BoundarySegment_ID (Figure 4) and selects BoundarySegment_ID values that differ for the same values of TIN_ID. In the discussed example (in Section 3.1.1), 1029 TIN edges were selected from the overall set of 3374 TIN edges. In the next step, TIN edges that touch base points were removed from the set of the selected TIN edges.

TIN_PointTopology_Conect_to_BoundarySegment

| POINT_ID | Geometry | TIN_ID | BoundarySegment_ID |
|---|---|---|---|
| 733 | Point | 367 | 9 |
| 734 | Point | 367 | 9 |
| 735 | Point | 368 | 9 |
| 736 | Point | 368 | 9 |
| 737 | Point | 369 | 9 |
| 738 | Point | 369 | 9 |
| 739 | Point | 370 | 10 |
| 740 | Point | 370 | 9 |
| 741 | Point | 371 | 9 |
| 742 | Point | 371 | 10 |
| 743 | Point | 372 | 9 |
| 744 | Point | 372 | 10 |
| 745 | Point | 373 | 10 |
| 746 | Point | 373 | 10 |
| 747 | Point | 374 | 9 |
| 748 | Point | 374 | 10 |
| 749 | Point | 375 | 10 |
| 750 | Point | 375 | 10 |
| 751 | Point | 376 | 9 |
| 752 | Point | 376 | 9 |

| POINT_ID | Geometry | TIN_ID | BoderSegment_ID | Comparison |
|---|---|---|---|---|
| 733 | Point | 367 | 9 | |
| 734 | Point | 367 | 9 | TRUTH |
| 735 | Point | 368 | 9 | |
| 736 | Point | 368 | 9 | TRUTH |
| 737 | Point | 369 | 9 | |
| 738 | Point | 369 | 9 | TRUTH |
| 739 | Point | 370 | 10 | |
| 740 | Point | **370** | 9 | **FALSE** |
| 741 | Point | 371 | 9 | |
| 742 | Point | **371** | 10 | **FALSE** |
| 743 | Point | 372 | 9 | |
| 744 | Point | **372** | 10 | **FALSE** |
| 745 | Point | 373 | 10 | |
| 746 | Point | 373 | 10 | TRUTH |
| 747 | Point | 374 | 9 | |
| 748 | Point | **374** | 10 | **FALSE** |
| 749 | Point | 375 | 10 | |
| 750 | Point | 375 | 10 | TRUTH |
| 751 | Point | 376 | 9 | |
| 752 | Point | 376 | 9 | TRUTH |

(a) (b)

**Figure 4.** Selection of TIN edges (TIN_ID) based o: (**a**) attributes of TIN vertices based on the topological relations during data processing; (**b**) ID of TIN edges selected based on FALSE values (370, 371, 372, 374, etc.).

### 2.1.2. Generation of a Skeleton Based on Segments of a Complex Polygon, Base Points and Midpoints of the Selected TIN Edges

The skeleton was generated with dedicated algorithms implemented by the authors in Python based on the GDAL library and own modifications. The proposed algorithm offers a simple solution: a skeleton of a polygon is generated after unimportant TIN edges and edges that touch base points have been eliminated. The input data were the segments of a complex polygon and a set of points (base points and midpoints of the selected TIN edges). The developed tool for generating and smoothing the skeleton is presented in the data and software repository.

### 2.1.3. A Comparison of BPSplit and Splitarea Algorithms

In 2016, Meijers et al. [10] proposed the Splitarea algorithm for generating skeletons of polygons that represent water bodies. In this study, the Splitarea algorithm was adopted as a classical method. The cited study inspired the authors to conduct the present research. The Base Point Split (BPSplit) algorithm was described by comparing it with the Splitarea algorithm [10]. Successive stages of the skeleton generation process involving the compared algorithms are presented in Table 1 and Figure 5. The presentation relies on the polygons developed by Meijets et al. [10]. Successive stages of the skeleton generation process involving the compared algorithms are presented in Table 1 and Figure 5.

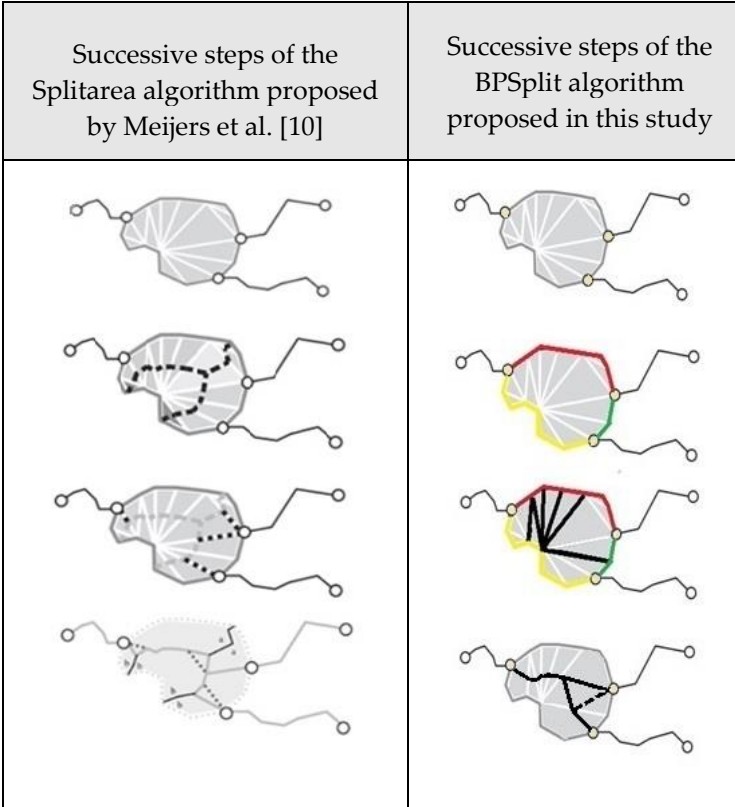

**Figure 5.** Successive steps of the algorithm generation process with the use of Splitarea [10] and BPSplit algorithms. The operations involving Table 1 data were visualized.

Successive data processing steps are presented in Figure 5. In the solution proposed by Meijets et al. [10], a skeleton is generated based on all TIN edges, and it is modified in successive steps to deliver the required functionality. In the approach involving the BPSplit algorithm, a set of base points is generated. In the presented example, base points are represented by three points on the polygon boundary line. Base points were used to divide the polygon boundary into segments, and the segments

were used to select TIN edges that touch different segments of the boundary lines, but not base points. A skeleton was generated based on the midpoints of the selected TIN edges and base points.

**Table 1.** Process of generating a skeleton with the Splitarea algorithm and the Base Point Split algorithm.

|  | **Classical Algorithm: Splitarea [10]** | **Proposed Algorithm: BPSplit** |
|---|---|---|
| (1) | Triangulation | Segmentation of polygon boundaries with the use of base points |
| (2) | Selection of internal triangles | Triangulation |
| (3) | Skeleton generation | Selection of internal triangles |
| (4) | Generation of connectors | Selection of TIN edges based on segments of the polygon boundary |
| (5) | Edge labeling and skeleton pruning | Generation of the final skeleton |
| (6) | Generation of the final skeleton | Skeleton smoothing |

### 2.1.4. Skeleton Adjustment with the BPSplit Algorithm

The skeleton generated with the BPSplit algorithm can contain loops (Figures 6 and 7). The resulting skeleton can be used to model navigation networks. The modeled hydrographic network should not contain loops. There are two approaches to removing loops. In the first method, a loop is replaced with a loop centroid (Figure 6b,d). This solution has been used in the literature for type 3 triangles (Figure 1). The second approach involves a tool for generating the shortest path between the source and the mouth of a river (the spanning tree tool can also be used). The results produced by the shortest path tool are combined to generate a skeleton without loops (Figure 6c).

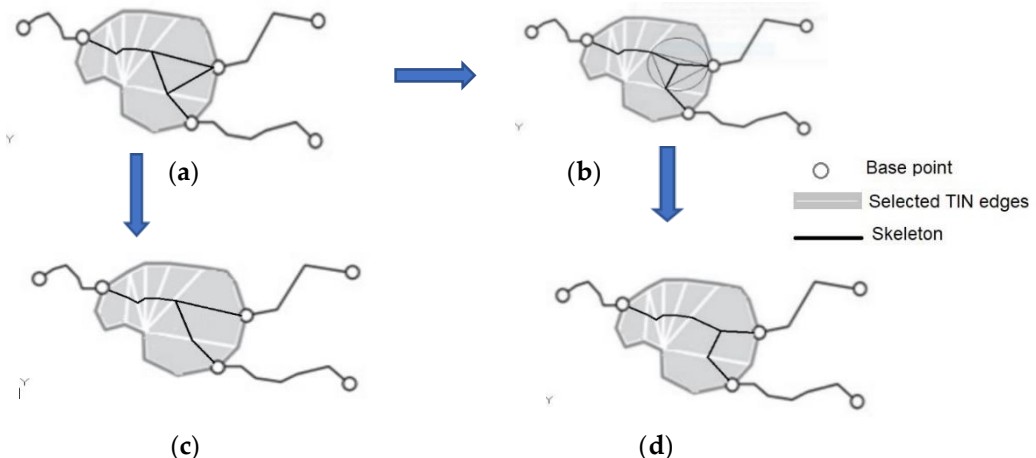

**Figure 6.** Elimination of loops from a skeleton: (**a**) skeleton with a loop; (**b**) the loop is replaced with the loop centroid (type 3 triangle); (**c**) the skeleton is modified with the use of network analysis tools; (**d**) the resulting skeleton where the loop was replaced with the loop centroid.

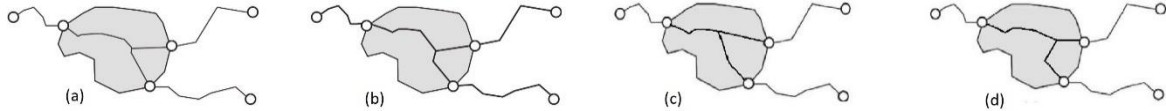

**Figure 7.** Skeletons of the same polygon generated with the (**a**) straight skeleton method [6]; (**b**) Splitarea algorithm [10]; (**c**,**d**) BPSplit algorithm and skeleton adjustment.

The skeletons generated with the use of four different methods are presented in Figure 7. The skeletons generated with the use of the solutions proposed by [10] are presented in Figure 7a,b. The skeletons generated with the use of the solutions proposed in this study are presented in Figure 7c,d.

The results presented in Figure 7 are different. The proposed solution in Figure 7c has the smallest number of edges. There are no good and bad solutions because every solution is correct. However, the multiplicity of the existing algorithms suggests that there is no single ideal solution. The applicability of the proposed solution should be evaluated by the users. The total length of skeleton edges can be compared. The skeleton's fit to river lines can be evaluated, but this approach requires additional hydrographic analyses. The complexities of the algorithm shown in Table 1 and Figure 5 can be compared. The BPSplit algorithm is simpler than the Splitarea algorithm.

### 2.1.5. Summary of the BPSplit Algorithm

The BPSplit algorithm generates a skeleton of a polygon with the use of base points and the midpoints of the selected TIN edges. The algorithm relies on various types of polygon segments (Figure 8a) and different connections with base points (Figure 8b) in the process of generating a skeleton.

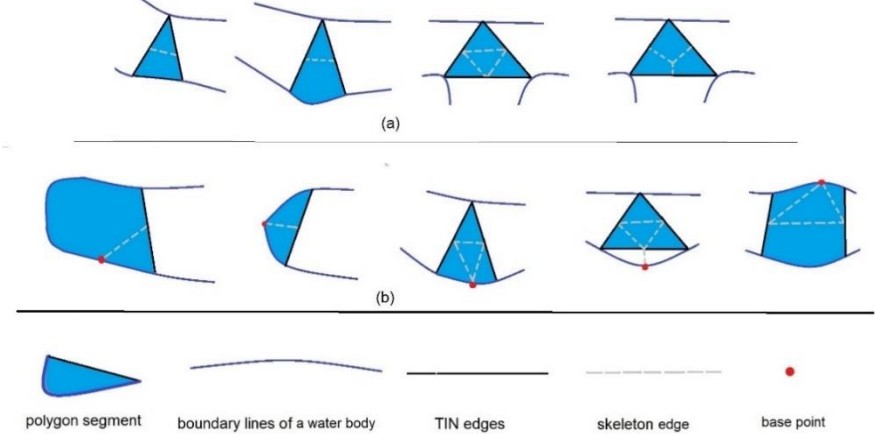

**Figure 8.** Editing of skeleton edges based on various types of polygon segments: (**a**) different types of polygon segments with skeleton edges; (**b**) different locations of base points in segments with a visualization of skeleton edges.

The proposed BPSplit algorithm relies on classical skeleton generation methods that are based on TIN edges. The relations between polygon boundaries and the neighboring polygons on the right and left side are not taken into account. A skeleton is generated based on the rules applicable to the identified types of polygon segments, as well as additional rules, with different locations of base points (Figure 8).

### 2.2. Methodology Associated with the Second Research Task—Polygons with Holes

The second research task relied on the assumption that the location of a skeleton representing a polygon with numerous holes (islands on water bodies) can be automatically modified between islands according to the preset criteria. In the selected software solutions, including ESRI, holes are eliminated during skeleton generation. This is the simplest solution. In the approach described by Meijers et al. [10], holes were taken into account, and their particular location inside the polygon was considered (Figure 9(a1,a2)). The special case where the holes inside a polygon come into contact is presented in Figure 9(a2). This solution was also inspired by the work of Meijers et al. [10]. The data for the BPSplit algorithm (Figure 9(b1,b2)) were prepared based on the complex polygons presented in [10] (Figure 9(a1,a2)).

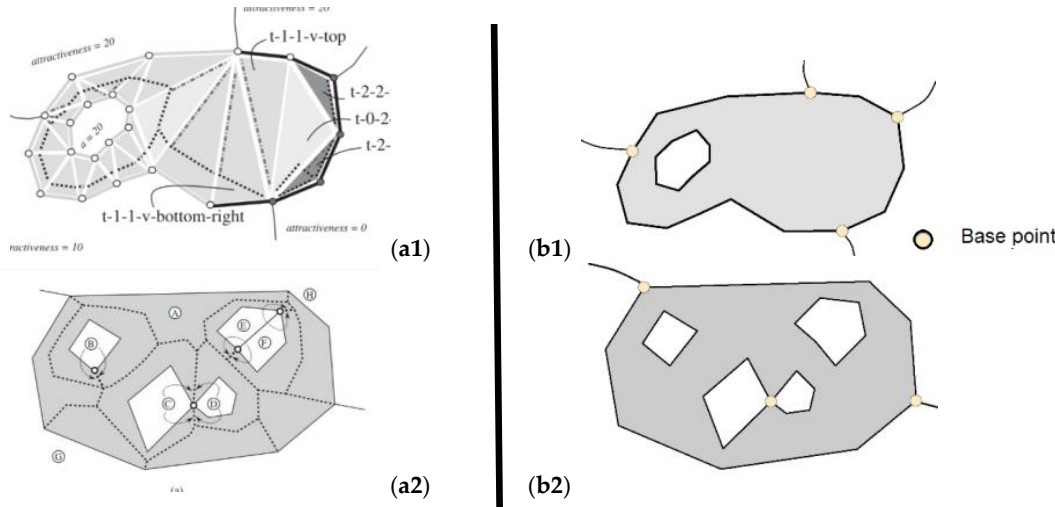

**Figure 9.** Complex polygons with holes: (**a1**,**a2**) are based on the work of Meijers et al. [10]; (**b1**,**b2**) represent complex polygons with one hole and numerous holes for testing.

Skeletons were generated with the use of the BPSplit algorithm based on the complex polygons presented by [10]. In the first step, base points were identified as common points located on both the polygon boundary and the lines touching the polygon contour. Four base points were identified for the complex polygon in Figure 8b. Three base points were identified for the polygon in Figure 9(b2), but two points were located on the polygon's external boundary, whereas the third point was located on the boundary of two contacting holes (the third point was adopted to generate skeleton edges between islands).

The set of base points was used to segment the boundaries of a complex polygon. The segments of polygon boundaries are marked with different colors in Figure 10. The boundaries of holes inside the polygon constitute separate segments. Polygon skeletons were generated with the BPSplit algorithm. TIN edges were generated based on the vertices of a complex polygon as well as the midpoints of polygon edges. These points had to be added to the boundary lines, because in preliminary tests where points were not added, individual edges of the skeletons generated with the BPSplit method intersected islands. This result points to an insufficient number of points on the polygon boundary for generating TIN edges with the BPSplit algorithm. In ESRI software [21], points are clustered by default when the centerline of a polygon is generated with the use of the Polygon to Centerline tool.

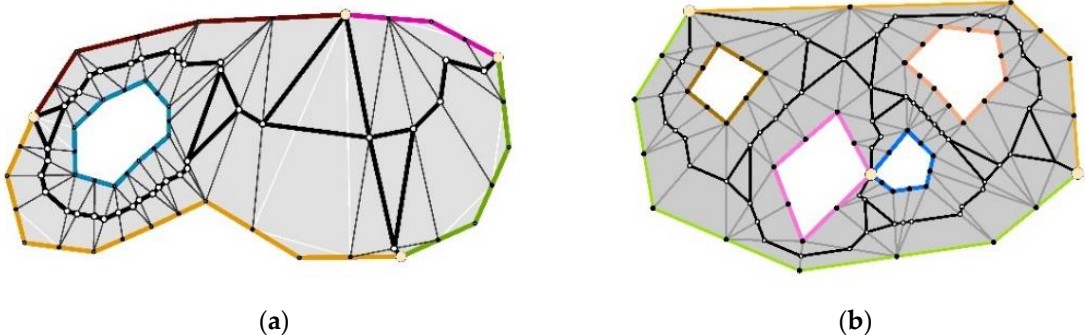

(**a**)                                                                                  (**b**)

**Figure 10.** Skeletons generated by the BPSplit algorithm based on the spatial data presented in Figure 9: (**a**) skeleton of a polygon with a single hole; (**b**) skeleton of a polygon with four holes.

The skeletons generated by the BPSplit algorithm are presented in Figure 10a,b. The skeletons surround the holes inside the polygon, and they contain loops. Loops can be replaced with loop centroids based on the rule applicable to type 3 triangles (Figures 1, 3d and 6a).

In the second research task, the authors assumed that skeletons which account for all holes inside polygons can be modified according to need. A diagram of an algorithm for modifying a skeleton is presented in Figure 11.

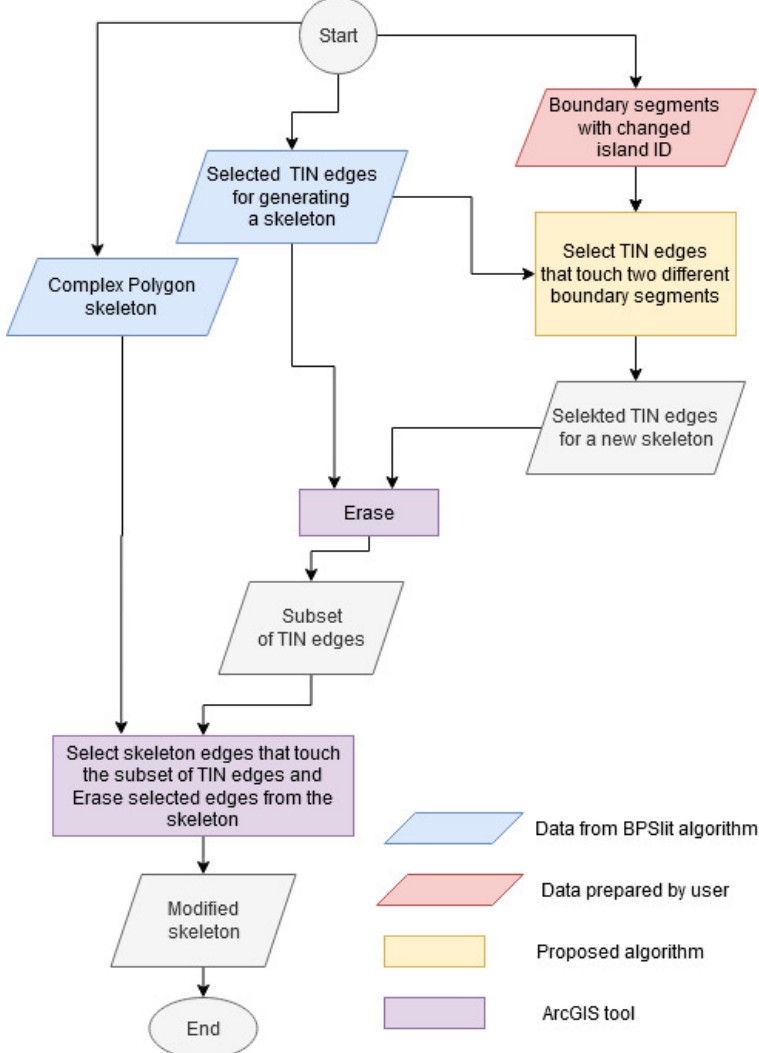

**Figure 11.** Flowchart of the algorithm which modifies a polygon skeleton by changing island identifiers.

The process of modifying a skeleton will be presented on the example of the skeletons shown in Figure 12.

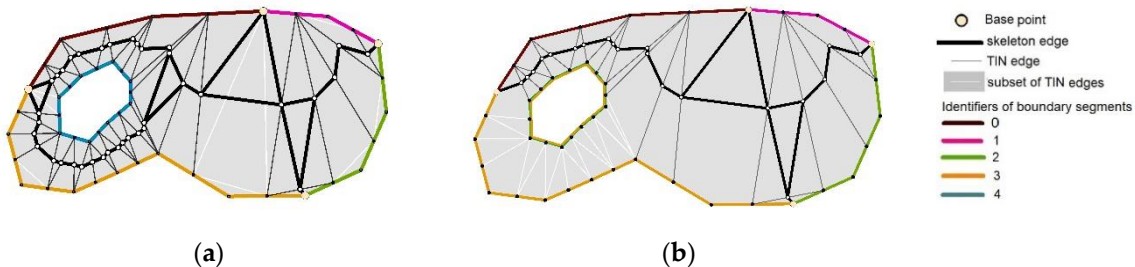

(**a**)                                                                    (**b**)

**Figure 12.** Skeletons generated for a polygon with a hole: (**a**) skeleton generated based on source data; (**b**) skeleton modified on the assumption that the skeleton should not be located south of the hole (the island ID was modified by changing the color of the island boundary).

The BPSplit algorithm generates a skeleton based on the midpoints of the selected TIN edges. The selected TIN edges have to touch two different polygon boundary segments. A polygon with a single hole features segments of the external boundary line and the hole boundary line which is regarded as the successive segment. If the ID of the hole boundary is replaced with the ID of a single segment of the external boundary, a skeleton will not be generated between the hole and that segment of the external boundary. This solution was applied to a polygon with a single hole (Figure 10a). It was assumed that a skeleton should not be located south of the hole. Therefore, the hole ID was replaced with the ID of the segment of the external boundary located south of the hole. As a result, the hole and the segment of the external boundary were assigned identical IDs. TIN edges that touch the hole and the analyzed segment of the external boundary were not considered in the process of selecting TIN edges. A subset of TIN edges can be created by comparing the selection of TIN edges with the BPSplit algorithm based on source data (Figure 12a) and the selection of TIN edges based on the modified ID of the hole boundary line (Figure 12b). The subset of TIN edges is the difference between TIN sets. To modify the skeleton, the skeleton edges that touch TIN edges from the above subset of TIN edges were eliminated. The skeleton generated based on source data is presented in Figure 12a, and the modified skeleton is shown in Figure 12b. Changes in the color of the hole boundary and the subset of TIN edges for modifying the skeleton are presented in Figure 12b.

Other solutions can be applied to complex polygons containing a large number of holes (Figure 10b):

- To automate the process, each hole boundary (island boundary) can be assigned an ID corresponding to the nearest segment of the external polygon boundary (based on hole centroids). TIN edges between holes and the external boundary lines will be disregarded in the process of selecting TIN edges. The edges of the modified polygon will be located between holes, on the polygon's centerline (Figure 13a).
- If a single ID is assigned to all holes, TIN edges between holes will not be considered during the automatic selection of TIN edges (Figure 13b). The edges of the skeleton will be located near two external polygon boundaries (Figure 13b).
- The ID of a selected segment of the external polygon boundary can be assigned to the boundary lines of the selected holes (islands). TIN edges between the selected holes are not considered. TIN edges between the selected holes and the selected segments of the external boundary are not considered either (Figure 13c).

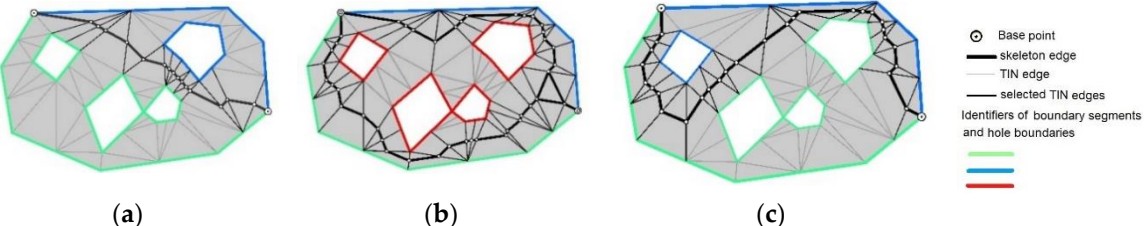

|     (**a**)     |     (**b**)     |     (**c**)     |

**Figure 13.** Skeleton modification based on three criteria: (**a**) the skeleton should be located in the center of the complex polygon between holes (islands); (**b**) the skeleton should omit holes (islands) and should be located along the external boundary of the complex polygon; (**c**) the skeleton can also be modified by changing the ID of hole boundaries.

The third research task is presented in Section 3.3.3, based on the discussed example. A flowchart of the algorithm for skeleton generalization is presented on Figure 14.

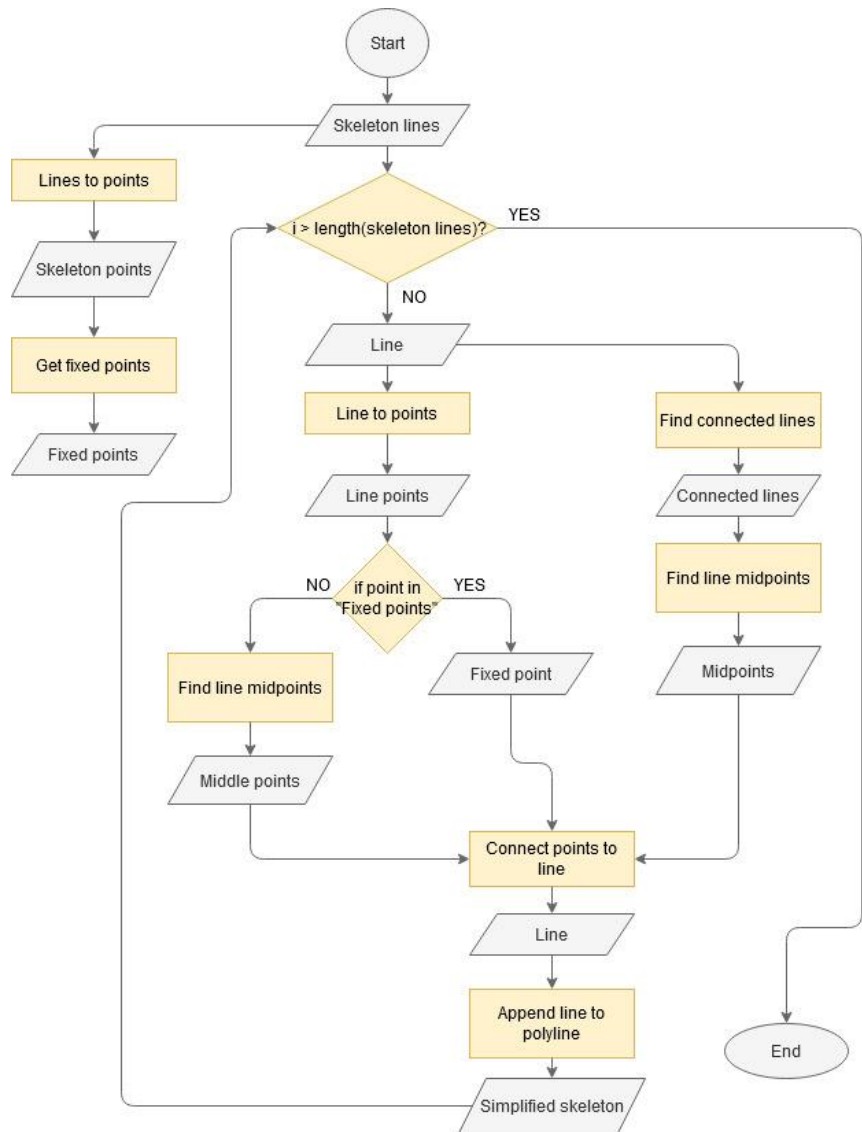

**Figure 14.** A flowchart of the algorithm for skeleton generalization.

## 3. Validation of the Algorithm on Large Data Sets

The BPSplit algorithm was applied to generate a skeleton of a hydrographic network based on vector data from the Database of Topographic Objects in 1:10,000 scale (BDOT_10). The database was created by public administration services based on aerial images. Data concerning lakes and rivers were selected from the database, and they were used to present the operation of the BPSplit algorithm.

### 3.1. The Application of the BPSplit Algorithm to Different Sets of Base Points in a Selected Object

In the first example, the algorithm was applied to generate a skeleton of a hydrographic network comprising two large lakes and two small lakes connected by a river, as well as seasonal inflows to the largest lakes (Figure 15). All lakes are represented by complex polygons. The river flows through all four lakes. The river and seasonal streams are represented by lines.

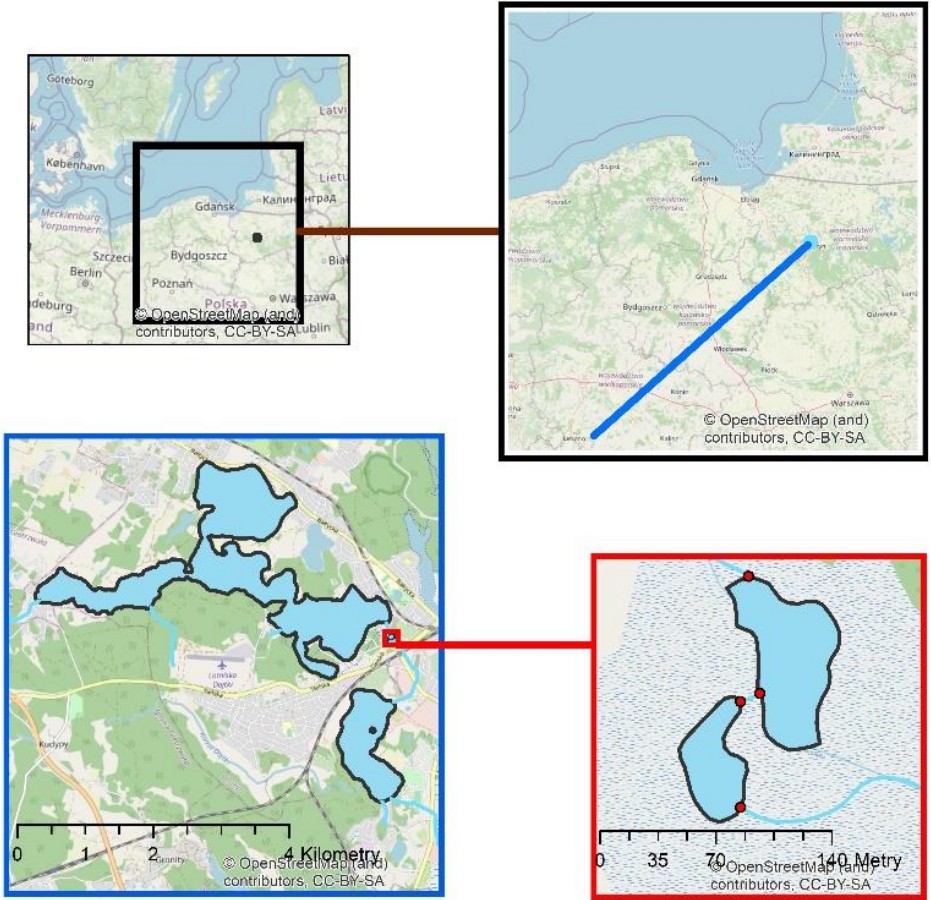

**Figure 15.** The analyzed object—four lakes connected by a river with seasonal streams flowing into the lakes.

The skeleton of the hydrographic network was generated based on source data in the following steps:

1.  A set of base points representing river inflows and outflows in lakes was created.
2.  The set of base points was used to divide the boundaries of complex polygons.
3.  TIN edges were generated inside complex polygons based on polygon vertices and base points.
4.  TIN edges that touch different segments of complex polygon boundaries, but do not touch base points, were selected.
5.  The midpoints of the selected TIN edges were generated.
6.  Complex polygons were divided into segments.
7.  Skeleton edges were generated between the midpoints of the selected TIN edges, and between the midpoints of TIN edges and base points.

The modeled hydrographic network consists of complex polygon skeletons and river lines.

Two solutions were applied to generate a skeleton of the hydrographic network. In the first solution, the skeleton was created based on the examined lakes and the river flowing through the lakes. Eight base points were generated to indicate the locations where the river flows in and out of the analyzed lakes. Seasonal streams entering the examined lakes were considered in the second solution, and eleven base points were generated.

In the third solution, the skeleton of the largest lake was modeled for the purpose of navigation. Attractive points on the lake shore were selected as the base points. The solutions was presenter in next sections

### 3.1.1. Generation of a Skeleton of a Hydrographic Network Based on Lakes and One River

In the first solution, the Kortówka River that flows through the four analyzed lakes was represented with a line. Eight base points marking the locations were the river flows in and out of the studied lakes were selected. Base points were used to divide lake boundaries into nine segments, including one segment representing the island boundary (Figure 16).

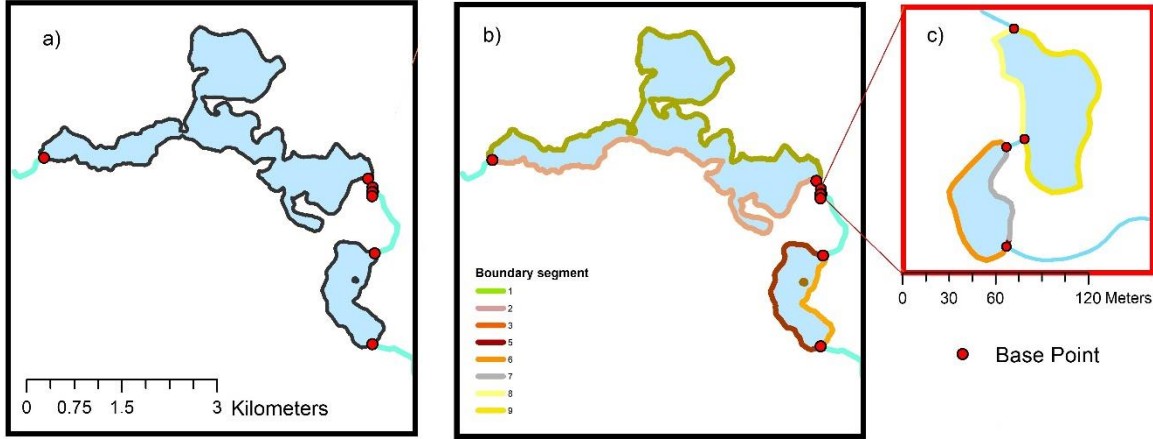

**Figure 16.** Source data for generating the first skeleton based on 8 base points and 9 segments of polygon (lake) boundaries: (**a**) lakes with the location of base points; (**b**) segmentation of the boundary lines of lakes; (**c**) two small lakes with segments of their boundary lines.

Boundary segments were used to select TIN edges which play an important role in the process of developing skeletons. The skeletons of lake polygons were generated based on the midpoints of the selected TIN edges and the set of base points (Figure 17).

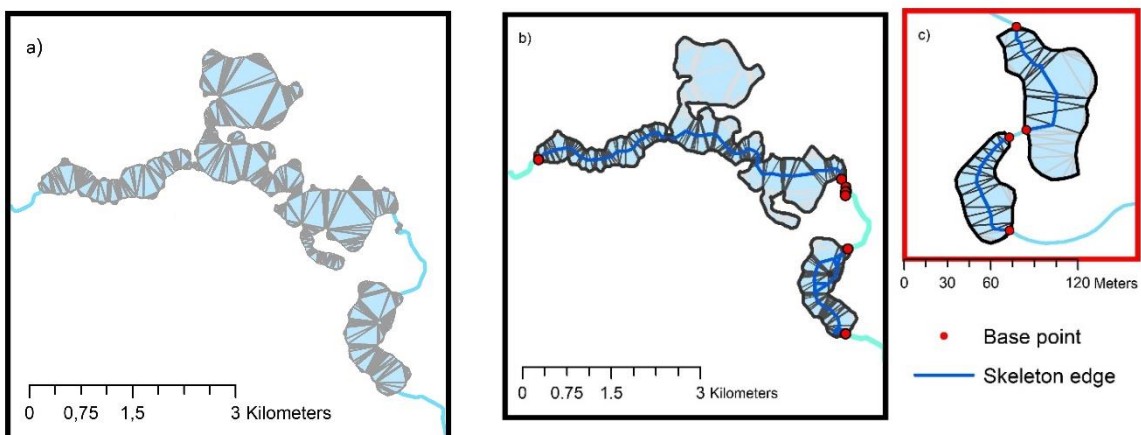

**Figure 17.** Skeleton of a hydrographic network generated with the BPSplit algorithm based on the selected base points: (**a**) TIN edges in polygons (lakes); (**b**) the selected TIN edges; (**c**) two small lakes with the selected TIN edges.

The network was generated based on polygon segments which were obtained by dividing polygons with the selected TIN edges (Figure 18). Only several segments had the shape of type 1 triangles. Type 1 triangles are encountered in narrow water bodies with a monotonous shore line, whereas type 3 triangles are found in the vicinity of islands. Skeleton lines were plotted in these segments (triangles) according to the rule presented in Figure 8.

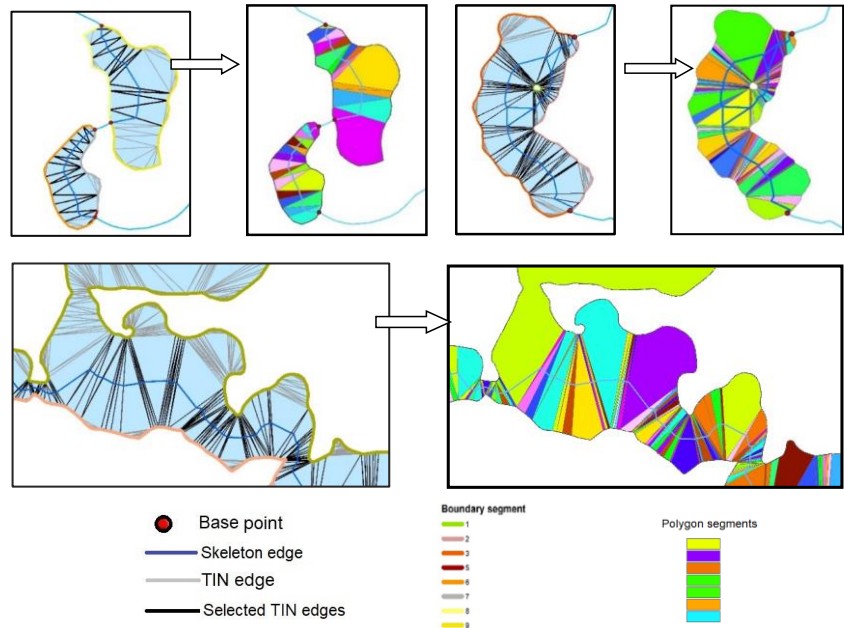

**Figure 18.** Examples of different polygon segments for generating a skeleton based on the midpoints of the selected TIN edges. A few segments have the shape of type 1 and 3 triangles from Figure 8.

### 3.1.2. Generation of a Skeleton of a Hydrographic Network Based on Lakes, One River and Three Streams

In the second solution, the main river intersecting all of the analyzed lakes and three seasonal streams flowing into these lakes were taken into account in the process of generating a skeleton. The number of base points increased by three, relative to the previous solution. These points were added to the boundaries of polygons representing the two largest lakes. The locations of the base points, selected TIN edges, skeletons generated on lakes, and seasonal streams are presented in Figure 19. Geometric figures in boundary segments touching base points are shown in Figure 20.

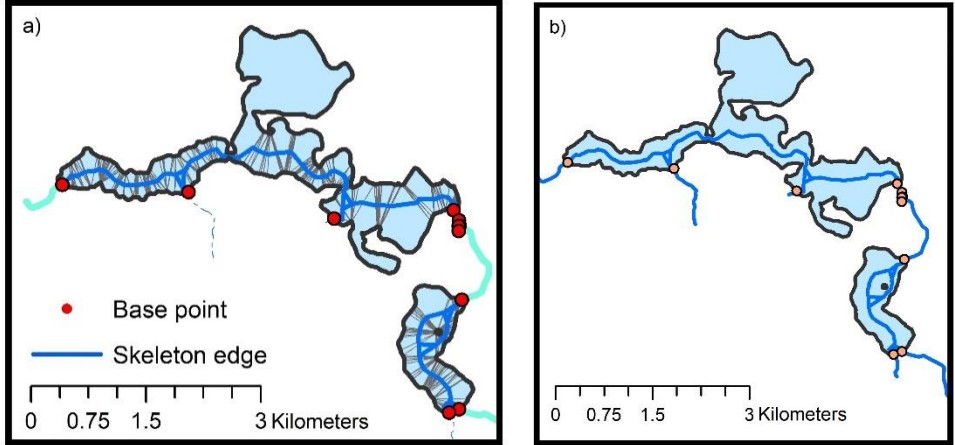

**Figure 19.** Skeleton of the hydrographic network generated based on 11 base points in four lakes, one river intersecting all lakes, and three seasonal streams flowing into the analyzed lakes: (**a**) generation process; (**b**) results.

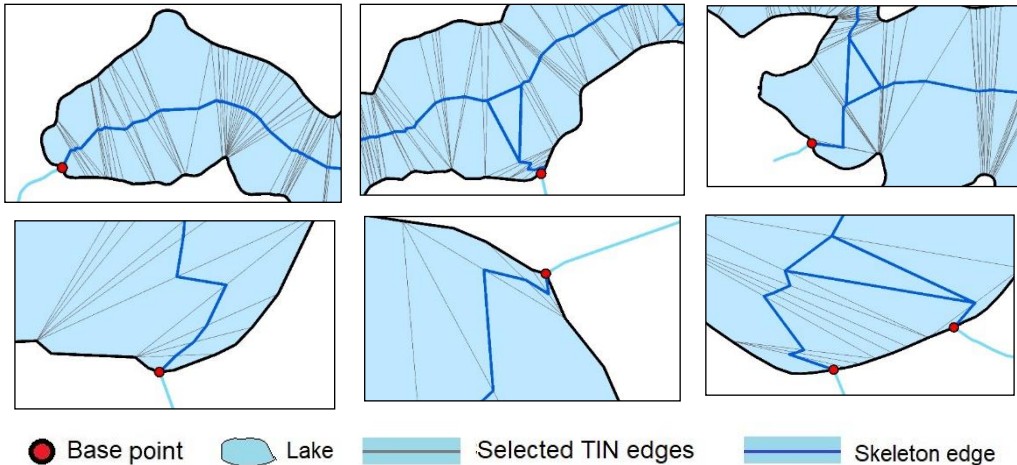

**Figure 20.** Different geometric figures in boundary segments near base points.

### 3.1.3. Generation of a Skeleton for Navigation in the Largest Lake with a Diverse Shore Line

The third solution involves only the largest lake as well as base points marking the most attractive sites on the lake shore. The generated skeleton can be used for navigation (Figure 21).

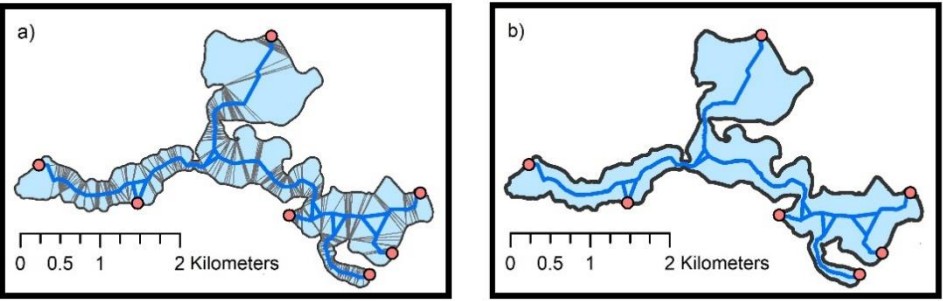

**Figure 21.** Skeleton of a polygon generated based on 7 base points marking attractive sites on the lake shore. The skeleton can be used for navigation; (**a**) generation process; (**b**) results.

### 3.2. The Results Generated by the BPSplit Algorithm on a Lake with Numerous Islands

In the next solution, the BPSplit algorithm was tested on a complex polygon containing many holes. The studied object was Lake Wydmińskie, which is characterized by a diverse shore line and numerous islands (Figure 22). A skeleton was generated for the purpose of navigation. Base points representing the launch point and mooring points were selected on the lake shore.

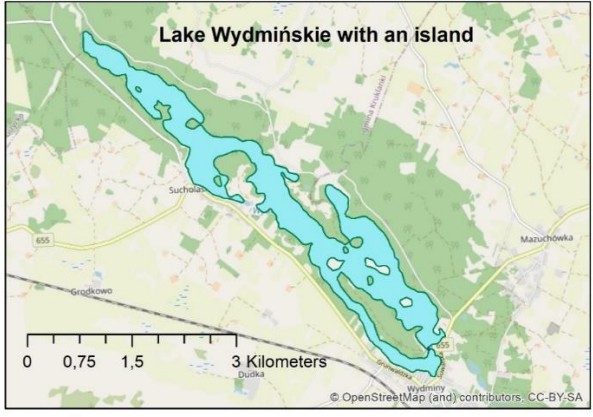

**Figure 22.** Lake Wydmińskie with numerous islands.

Data were processed with the BPSplit algorithm. The results of successive transformations are presented in Figure 23. The generated skeleton is shown in Figure 24.

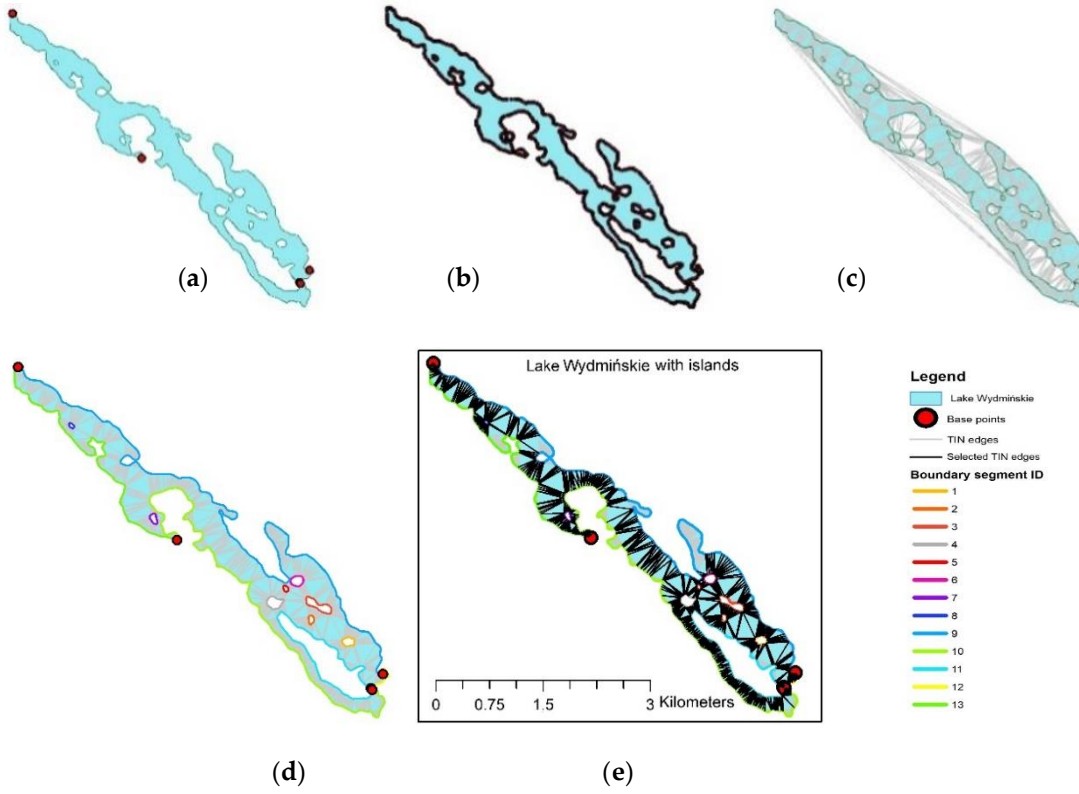

**Figure 23.** Data transformation in the skeleton generation process: (**a**) location of base points, (**b**) vertices of a complex polygon representing the analyzed lake; (**c**) generated TIN edges; (**d**) TIN edges inside the complex polygon representing the analyzed lake, with a visualization of boundary segments; (**e**) process visualization, with an indication of the TIN edges selected for skeleton generation.

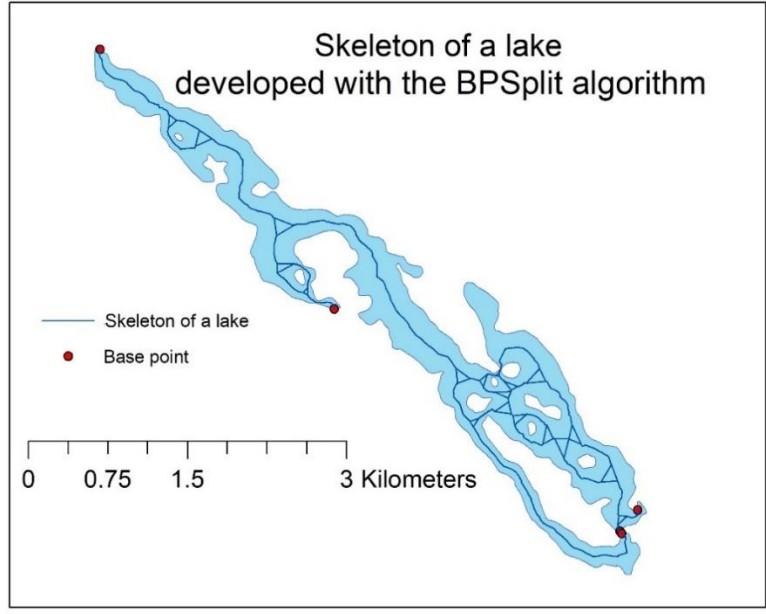

**Figure 24.** Skeleton of a lake with numerous islands developed with the BPSplit algorithm based on the adopted set of base points.

The presented skeleton was generated based on the location of base points, and it does not cover the entire lake (polygon). The skeleton accounts for significant areas around the selected base points. The skeleton covers all islands. Skeleton edges were generated on both sides of the islands.

### 3.3. Modification of a Skeleton Representing a Polygon with Many Holes—Validation of the Solution Proposed in the Second Research Task

The assumption that a skeleton between islands can be modified automatically was validated on the example of Lake Wydmińskie (Figure 24) as the base skeleton. The selected TIN edges for building the base skeleton and the selected TIN after the modification of the island ID were used in the modification process. Attention should be paid to the difference between these sets, i.e., a subset of TIN edges. The base skeleton was modified by eliminating skeleton edges that touch the edges of a subset of TIN.

Two modifications involving automatic changes in the ID of island boundaries are presented below. A diagram of the algorithm for modification of the skeleton by changing the ID of hole boundaries is presented in Figure 11.

### 3.3.1. Modification of the Base Skeleton to Generate a Skeleton between Islands in the Center of the Lake by Assigning the ID of the Nearest Segment of the Polygon Boundary to Island Boundaries

The base skeleton was modified by replacing the ID of island boundaries with the ID of the nearest segment of the external lake boundary. When the ID of island boundaries was modified, a new set of TIN edges touching different segments of the polygon boundary was selected. The base set of TIN edges was compared with the new set of TIN edges, and a subset of TIN edges was created based on the difference between the compared sets. The base skeleton was modified by removing the edges touching the subset of TIN edges. The modification process is presented in Figure 25. The new IDs of island boundaries are marked in color in Figure 25a. The subset of TIN edges for modifying the skeleton is presented in Figure 25b. The process of eliminating skeleton edges that touch subset of TIN edges is shown in Figure 25c. The resulting skeleton is presented in Figure 25d.

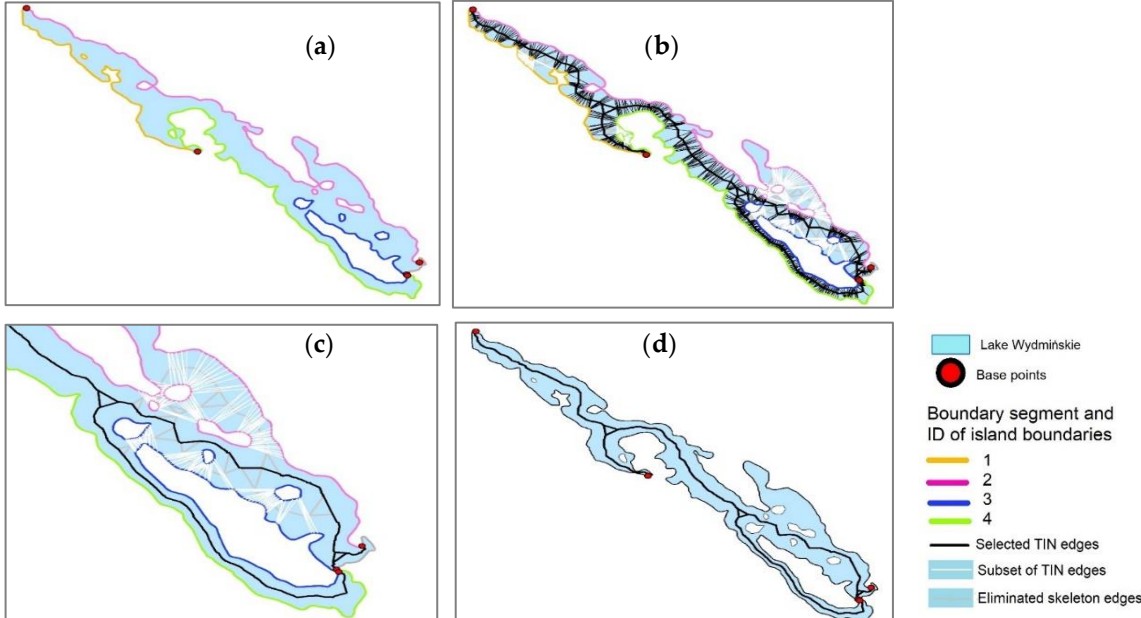

**Figure 25.** Modification of the base skeleton to obtain a skeleton in the center of the polygon between islands: (**a**) island boundaries are assigned new identifiers, and the resulting changes are marked in color; (**b**) a subset of TIN edges for modifying the skeleton is marked in white; (**c**) elimination of skeleton edges that touch TIN edges; (**d**) the resulting skeleton.

### 3.3.2. Modification of the Lake Skeleton to Obtain a Skeleton that Is Not Located between Islands When Island Boundaries Are Assigned an Identical ID

If all island boundaries are assigned an identical ID, the modified lake skeleton will be located along the lake's external boundaries, and it will not be located between islands. The successive steps of the skeleton modification procedure, during which the edges of the base skeleton touching the subset of TIN edges were removed, are presented in Figure 26.

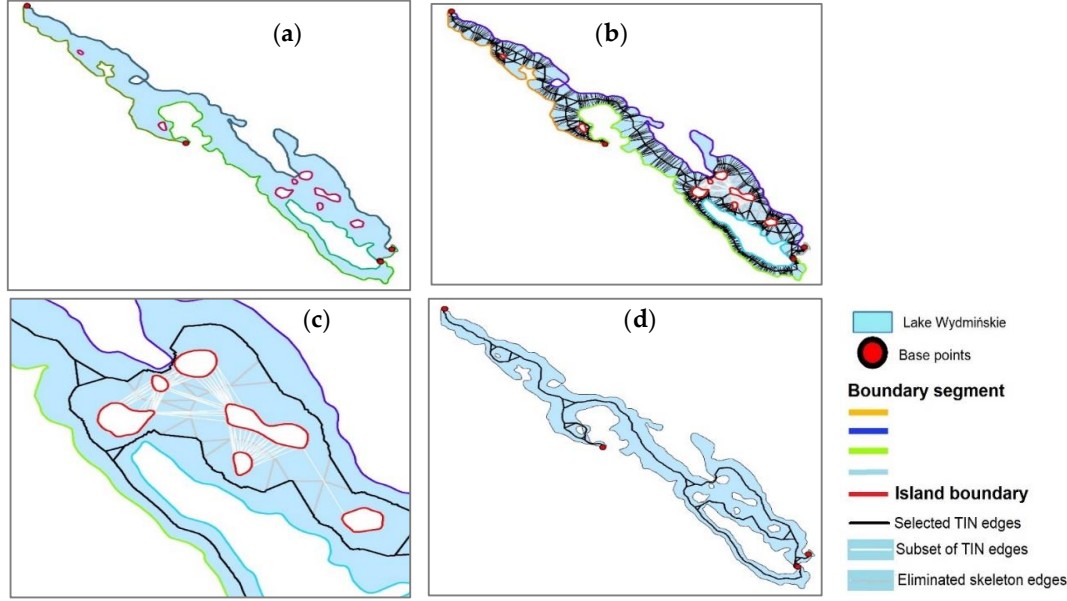

**Figure 26.** Modification of the base skeleton to obtain a skeleton located along the lake's external boundaries when the boundaries of all islands are assigned an identical ID: (**a**) the segments of island boundaries are assigned new identifiers, and the island boundary lines are marked with the same color; (**b**) the subset of TIN edges for modifying the skeleton is marked in white; (**c**) elimination of skeleton edges that touch the subset of TIN edges; (**d**) the resulting skeleton.

A base skeleton with holes can also be modified by changing the ID of island boundaries based on island attributes. The described procedures can be applied to predict the outcome of such modifications.

### 3.3.3. Skeleton Generalization (Smoothing)—Validation of the Third Research Task

In most solutions where skeletons are generated based on TIN edges, concave polygons and polygons with holes (islands) are characterized by the presence of loops and zigzag effects. The third research task was validated by smoothing the skeleton of Lake Wydmińskie (Figure 24). The algorithm is presented in the diagram in Figure 14. The loop was replaced with loop centroids (Figure 27) with the use of an algorithm developed by the authors in Python. A similar procedure was applied in the solution for type 3 triangles. The elimination of the loop reduces the number of nodes in the skeleton.

Most polygons generated based on TIN edges have zigzag effects. Zigzag effects are minimal when the polygon has uniform width, and they are more extensive in polygons with varied width (Figures 27c and 28a). Various line simplification methods are described in the literature [23–27]. An alternative smoothing method, which is a variant of the moving average approach, was proposed in this study. In this approach, skeleton segments are replaced by new segments based on the midpoints of successive skeleton segments (Figure 28b). To prevent the formation of new loops during smoothing, the points representing network nodes were adopted as fixed points. The set of base points was combined with the set of fixed points. It was assumed that the location of fixed points would not change during smoothing. Successive steps (iterations) of the smoothing procedure are presented in Figure 28c,d, and the result is shown in Figure 29.

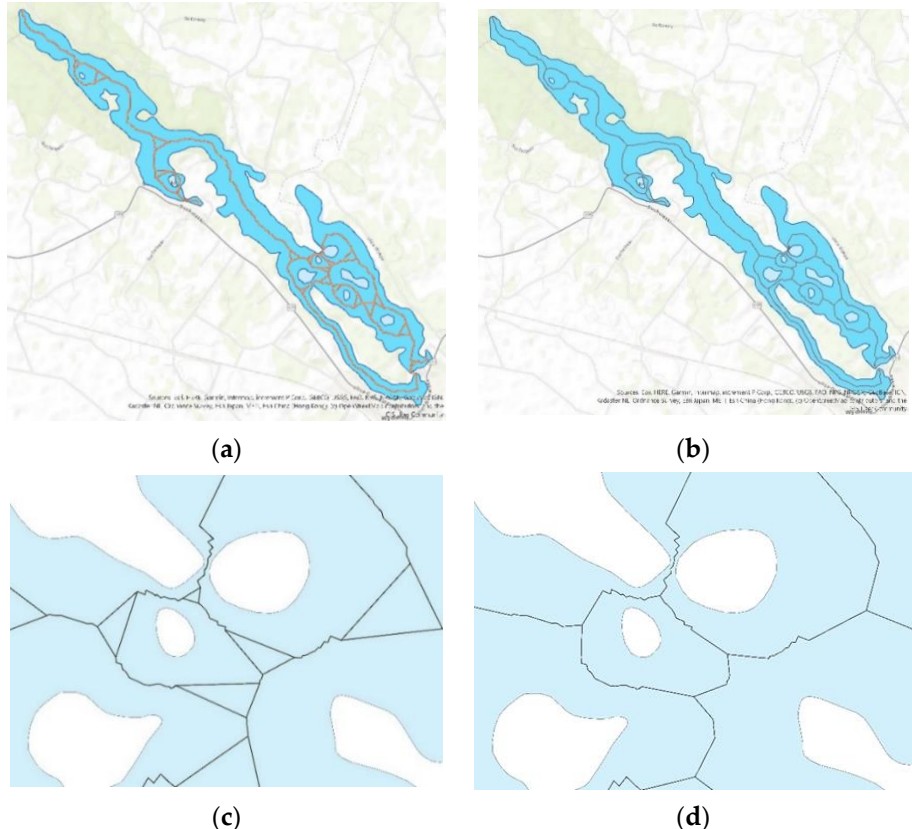

**Figure 27.** Skeleton model: (**a**) model with loops; (**b**) model where loops were replaced with loop centroids; (**c**,**d**) fragments of the skeleton before and after loop removal.

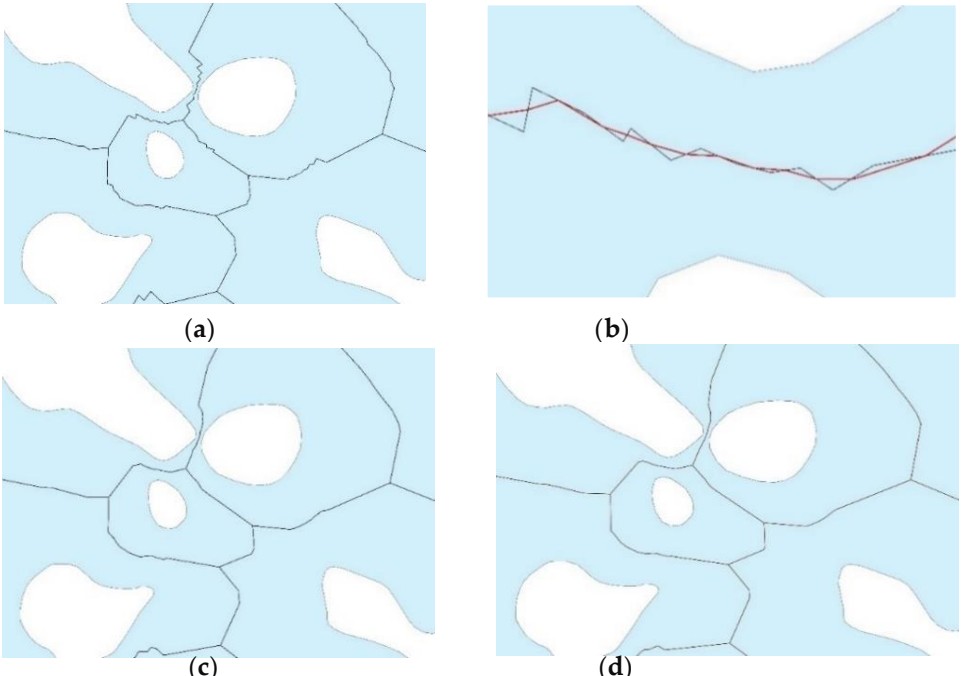

**Figure 28.** Elimination of zigzag effects in the network skeleton by establishing fixed points at network nodes: (**a**) zigzag effects in the network; (**b**) visualization of the smoothing method; (**c**) results of smoothing after the first iteration; (**d**) results of smoothing after the second iteration.

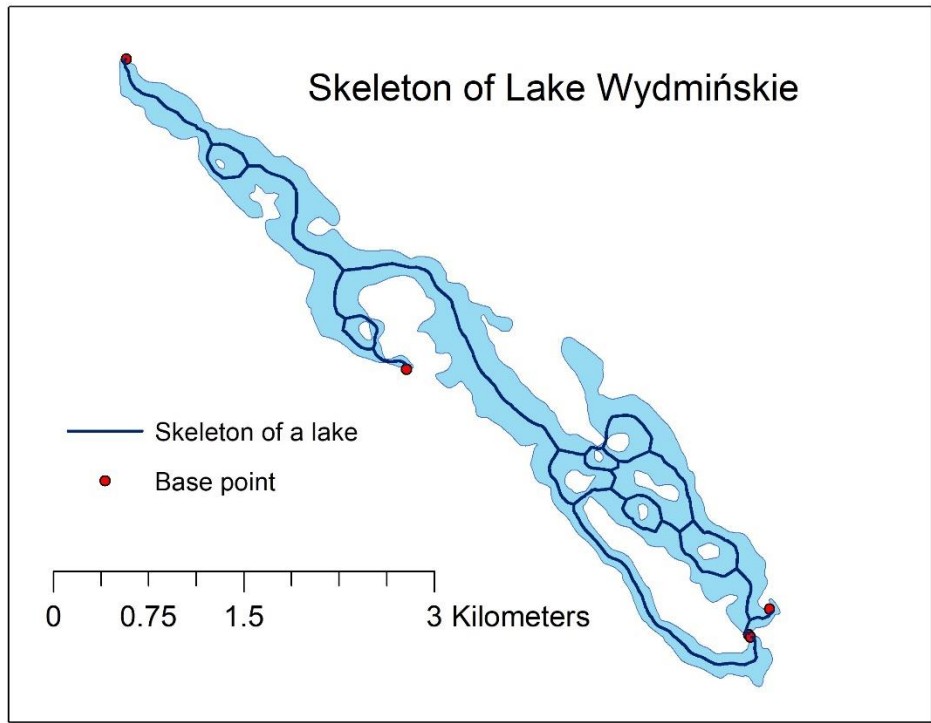

**Figure 29.** A model of the network skeleton generated in a geometrically complex polygon representing Lake Wydmińskie based on the adopted set of base points, after loop elimination and two iterations of the smoothing procedure to remove zigzag effects.

In successive iterations of the smoothing process, the location of the resulting object should be controlled inside the polygon.

## 4. Conclusions

This article proposes an improved version of the previously developed algorithm for generating skeletons of complex polygons [14]. The presented solution was inspired by the work of Meijers et al. [10].

The name of the BPSplit algorithm suggests that base points play an important role in the skeleton generation process. The selected base points on the polygon's boundary line guarantee that the skeleton will touch the polygon's boundary. These base points will be the end nodes of the skeleton's hanging edges.

In the developed algorithm, the proposed solutions for identifying base points, selecting important TIN edges for generating a skeleton, and modifying skeleton edges between islands seem noteworthy on account of their simplicity. These solutions constitute the new elements in the improved version of the previously presented algorithm [14].

Therefore, the proposed BPSplit algorithm is an alternative method of generating polygon skeletons. The algorithm can be applied to various sets of geospatial vector data, and to specific cases. The BPSplit algorithm for generating a polygon skeleton requires a set of base points on the polygon boundary line. Base points determine the shape of the generated skeleton, and they play an important role during the selection of TIN edges. TIN edges for skeleton generation are selected based on simple topological relations that are identified during simple data processing. The set of base points supports the determination of initial conditions in the process of generating a skeleton.

The proposed algorithm was validated on data sets with varied geometry, based on different sets of base points. The algorithm validated the first research task postulating that the anticipated shape of the generated skeleton can be defined.

In lakes with numerous islands (polygons with numerous holes), the BPSplit algorithm generates skeleton edges between islands, and between islands and external lake boundaries. The location of the

skeleton between islands can be modified by the user. The presented examples validate the second research task.

The third research task was also successfully validated by removing loops and smoothing the skeleton based on the defined nodes and base points as fixed points.

Several tests using different types of polygons shapes were presented. The results of the BPSplit algorithm are comparable with other solutions. The simplicity of the solution encourages further research in this field.

**Author Contributions:** Conceptualization, Elżbieta Lewandowicz; methodology, Elżbieta Lewandowicz; software, Elżbieta Lewandowicz; software by Python, Paweł Flisek; validation, Elżbieta Lewandowicz; formal analysis, Elżbieta Lewandowicz; writing—original draft preparation, Elżbieta Lewandowicz; writing—review and editing, Elżbieta Lewandowicz; visualization, Elżbieta Lewandowicz. All authors have read and agreed to the published version of the manuscript.

**Funding:** This research was financed as part of a statutory research project of the Faculty of Goengineering of the University of Warmia and Mazury in Olsztyn, Poland, entitled "Geoinformation from the theoretical, analytical and practical perspective" (No. 29.610.008, timeline: 2017–2020).

**Acknowledgments:** The authors are grateful to the Regional Center for Geodetic and Cartographic Documentation in Olsztyn for providing access to the Database of Topographical Objects in 1:10,000 scale. The acquired data supported analyses in the tested objects.

**Conflicts of Interest:** The authors declare no conflict of interest.

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
