# Peer review of "Base Point Split Algorithm for Generating Polygon Skeleton Lines on the Example of Lakes"

_ijgi, doi:10.3390/ijgi9110680_

Round 1

Reviewer 1 Report

General comments

This seems to be a rather sound piece of work that was properly and successfully accomplished by the authors. Thus, congratulations on the work undertaken.

In an earlier stage of their work, the authors developed an algorithm to generate the skeleton (centreline) of vector-based polygons describing lakes and rivers. That algorithm revealed to be inadequate though to deal with more complex polygons from the geometric point of view – i.e. a polygon containing other polygon islands. Dealing with geometrically more complex polygons, containing numerous holes, lead to consider all neighbouring polygons and to take into account all topological relations between polygon boundaries, which is in fact a rather challenging task.

Authors seem to have succeeded in the development of a more adequate algorithm for purposes above. In fact, several tests were carried out, using different sorts of polygons shapes, and the algorithm seems to have proved to actually accomplish what it was designed for.

I just would like to draw authors’ attention to the following aspects.

  1. When authors simply refer to “a polygon” in order to actually mean polygons with islands/holes, I would rather refer in all instances to “geometrically complex polygons” (or simply “complex polygons”), otherwise it may not be clear enough to the reader what the problem in hands is. I know the aim of the study is clarified in lines 68 & 69, even though I’d suggest for authors to check this throughout the document and adjust this in all instances.
  2. Several figures (e.g. Figures 2, 3, 8, 11, 15, 17) include sketches in different colour, type, and width lines. If this fact has any meaning in terms of illustration purposes, an adequate legend must be included – like you did, for instance, in Figure 9 and others.
  3. Given that the paper is about the design and implementation of an algorithm, I suppose that it would beneficial to include flowchart(s) as an overview of the whole process implemented by the algorithms. As such, possibly, 3 flowcharts could be inserted, one at the very beginning of the description of each research task, in order to illustrate how they were tackled and how the algorithms work.
  4. Finally, it would actually possible to provide the reader with all the software packages implemented in Python – let´s say, through an online “data & software” repository?

Detailed comments

Please, refer to the attached PDF.

Author Response

Responses to Reviewers 1 Comments

Point 1: ” This seems to be a rather sound piece of work that was properly and successfully accomplished by the authors. Thus, congratulations on the work undertaken. In an earlier stage of their work, the authors developed an algorithm to generate the skeleton (centreline) of vector-based polygons describing lakes and rivers. That algorithm revealed to be inadequate though to deal with more complex polygons from the geometric point of view – i.e. a polygon containing other polygon islands. Dealing with geometrically more complex polygons, containing numerous holes, lead to consider all neighbouring polygons and to take into account all topological relations between polygon boundaries, which is in fact a rather challenging task. Authors seem to have succeeded in the development of a more adequate algorithm for purposes above. In fact, several tests were carried out, using different sorts of polygons shapes, and the algorithm seems to have proved to actually accomplish what it was designed for. I just would like to draw authors’ attention to the following aspects.  When authors simply refer to “a polygon” in order to actually mean polygons with islands/holes, I would rather refer in all instances to “geometrically complex polygons” (or simply “complex polygons”), otherwise it may not be clear enough to the reader what the problem in hands is. I know the aim of the study is clarified in lines 68 & 69, even though I’d suggest for authors to check this throughout the document and adjust this in all instances. Several figures (e.g. Figures 2, 3, 8, 11, 15, 17) include sketches in different colour, type, and width lines. If this fact has any meaning in terms of illustration purposes, an adequate legend must be included – like you did, for instance, in Figure 9 and others. 

Response 1: Thank you for the positive feedback and valuable remarks. As suggested by the Reviewer, the terminology was modified and additional information was provided in figure drawings.

Point 2: Given that the paper is about the design and implementation of an algorithm, I suppose that it would beneficial to include flowchart(s) as an overview of the whole process implemented by the algorithms. As such, possibly, 3 flowcharts could be inserted, one at the very beginning of the description of each research task, in order to illustrate how they were tackled and how the algorithms work.  Finally, it would actually possible to provide the reader with all the software packages implemented in Python – let´s say, through an online “data & software” repository?”

Response 2: A flowchart with mathematical formulas illustrating the process implemented by the algorithm was included in the revised manuscript.

The study relied mainly on ArcGIS software and the associated framework tools. The computational complexity of the proposed algorithm is difficult to describe because data were processed with GIS tools, whereas an external algorithm from the GDAL library was used in the ESRI application. The developed code for generating a skeleton based on polygon segments, base points and the midpoints of selected TIN edges, and the code for simplifying the skeleton will be presented in the data&software repository. 

The authors would like to thank Reviewer, for the time and effort taken to review this paper.

Reviewer 2 Report

The article "Base Point Split Algorithm for Generating Polygon Skeleton Lines" addresses an important topic. Unfortunately, the article is very hard to read:

  • The input data not mentioned. The article should describe the input data and input parameters of the new algorithm.
  • The article should describe the goal, i.e. what is the desired result of the algorithm? What is a good or a bad result? And how to compare them?
  • In the introduction (lines 40ff): if a boundary polygon is given, and if the goal is to triangulate the polygon, then why do the resulting triangles overlap the polygon boundary?
  • Do the mentioned "research tasks" in section 2 describe the _steps_ of the algorithm?!
  • What is a "base point"? I did not find a suitable definition.
  • ...

In summary, the topic is interesting and the new algorithm may be a good contribution, but the article needs edited to be more readable.

Author Response

Responses to Reviewers 2 Comments

Point 1: The article "Base Point Split Algorithm for Generating Polygon Skeleton Lines" addresses an important topic. Unfortunately, the article is very hard to read: The input data not mentioned. The article should describe the input data and input parameters of the new algorithm.     The article should describe the goal, i.e. what is the desired result of the algorithm? What is a good or a bad result? And how to compare them?  

Response 1: Thank you for the time and effort taken to review our paper. As suggested, the input data and the processes implemented by the algorithm were described in greater detail. However, the algorithm was developed by converting framework data with the use of GDAL tools in ArcGIS. The final part of the algorithm was developed with the use of self-designed programming tools in Python. The code was presented in the repository.

 Point 2:  In the introduction (lines 40ff): if a boundary polygon is given, and if the goal is to triangulate the polygon, then why do the resulting triangles overlap the polygon boundary?

Response 2: Thank you for this valuable observation. The relevant changes were made in the revised manuscript.

Point 3: Do the mentioned "research tasks" in section 2 describe the _steps_ of the algorithm?!     What is a "base point"? I did not find a suitable definition.

Response 3: A detailed definition of a base point was provided in the revised manuscript. A complex polygon and a set of base points are required to process data with the BPSplit algorithm.

Point 4: In summary, the topic is interesting and the new algorithm may be a good contribution, but the article needs edited to be more readable.”

Response 4: Thank for you the positive feedback. The paper was restructured, and additional information was provided according to the Reviewer’s suggestions.

The authors would like to thank Reviewer, for the time and effort taken to review this paper.

Reviewer 3 Report

The paper considers the problem of generating polygon skeletons which depend on topological relations with adjacent lines at so-called base points. The structure of the paper includes five chapters dedicated to introduction to the problem, materials and methods, results, skeleton generalization, and discussion. The topic of the paper is within the scope of IJGI journal.

The current state of the manuscript does not allow me to recommend it for publication because of some flaws both in methodology and representation. I summarize the main issues below:

  1. The necessity of such developments is not substantiated. The Introduction section includes a concise overview of existing solutions to skeleton generation. However, the problems and shortcomings of these algorithms are not analyzed. Therefore, it is not clear, what particular scientific gap you are trying to fulfill.

  2. Following the previous comment, it is unclear, what is the novelty of the presented research. What particular task in skeleton generation can be solved by your algorithm, which cannot be solved using the previous approaches? I mention that you refer to the Splitarea algorithm extensively. In what aspect does your approch overperform Splitarea?

  3. The paper seems to include three research tasks, but the two of these tasks are never formulated in clear and explicit manner. First, the authors (on page 2) speak about assumption that the designed algorithm can be used to generate an appropriately shaped skeleton already at the beginning of the process. The second task focuses on polygons with numerous holes (islands on water bodies), where the skeleton’s location between islands was modified automatically based on the preset criteria. So, the first statement is about the assumption, and the second statement is about the focus. Then, what are the tasks related to these statements? The following content of the paper does not bring a specific explanations on this, and the reader must infer them from description of the methodology and results.

  4. The main subject of the paper is so-called BSplit algorithm for generating the skeleton. Surprisingly, the algorithm is never presented as a sequence of actions, which allows it to be reproduced. The Table 2 contains some comparison between Splitarea dand BPSplit, but the most complicated stage — skeleton generation is not explained. The paragraph 2.1.3 "Description of the BPSplit algorithm" discusses what the algorithm generates (a skeleton), and on what it is relied (classical skeleton generation methods) or based (rules applied to identified types of polygon segments). Then, what is the full algorithm itself? You should provide the reader with sequence of steps: 1, 2, ... and so on — starting with the source data and ending with generated skeleton.

  5. The degree of automation of the method is questionable. It seems that selection of the strategy to skeletonize polygon with many holes (ID of the nearest segment vs. similar ID for all island boundaries) relies on user decision. The paper does not discuss the advantages and shortcomings of both approaches, as well as some rules to select between them using automated decision strategy.

  6. I feel that the smoothing task does not deserve so much attention in the paper. Despite the claim that some alternative smoothing method has breen developed, the descrived strategy is simply the variant of widely used moving average approach.

  7. The overall style of narrative is hard to consume, because of some flaws in logical sequence of explanations and unusual terminology. For example, the assumptions are presented instead of the tasks themselves. And the term 'apex' is used for what is usually called 'node' or 'vertex' in GIScience.

  8. The final section of the paper called Discussion is stylistically more like a Conclusion. However, the Conclusion should not only summarize the results, but also bring some light on what particular advancement your research has brought to scientific community, to highlight the novelty of your method. None of this is present there. And to formulate your novelty, you should first better explain the real problem to solve in Introduction (see Issues 1 and 2).

Author Response

Responses to Reviewers 3 Comments

Point 1: “The paper considers the problem of generating polygon skeletons which depend on topological relations with adjacent lines at so-called base points. The structure of the paper includes five chapters dedicated to introduction to the problem, materials and methods, results, skeleton generalization, and discussion. The topic of the paper is within the scope of IJGI journal.”

Response 1: Thank you for the positive feedback. The first part of the paper and the conclusions were restructured based on the Reviewer’s valuable suggestions.

Point 2: The current state of the manuscript does not allow me to recommend it for publication because of some flaws both in methodology and representation. I summarize the main issues below:

    The necessity of such developments is not substantiated. The Introduction section includes a concise overview of existing solutions to skeleton generation. However, the problems and shortcomings of these algorithms are not analyzed. Therefore, it is not clear, what particular scientific gap you are trying to fulfill.

Response 2: As noted in the Introduction, the present study is a continuation of the research described in the paper entitled “A Method for Generating the Centerline of an Elongated Polygon on the Example of a Watercourse” [14]. This fact was emphasized in the revised manuscript.

In the previous study, the authors concluded that the proposed algorithm should be tested and modified in analyses of other objects with varied shapes, in particular in networks composed of a large number of connected water bodies [14] (p. 17).  Such attempts have been made in the current study to eliminate the shortcomings of the previously described algorithm, where the neighborhood relations between polygon boundaries and the neighboring polygons had to be taken into account. This task proved to be particularly challenging in polygons containing numerous holes.

The proposed algorithm eliminates this problem. The current study presents an alternative approach to the existing solutions. As indicated in lines 13 and 14, the expected shape of the polygon skeleton is determined by a set of base points which are selected by the user. The importance of base points was emphasized and explained in greater detail in the revised manuscript. The previous study [14] contains an extensive review of the literature which was not duplicated in the present paper. 

  Point 3:  Following the previous comment, it is unclear, what is the novelty of the presented research. What particular task in skeleton generation can be solved by your algorithm, which cannot be solved using the previous approaches? I mention that you refer to the Splitarea algorithm extensively. In what aspect does your approch overperform Splitarea?

Response 3: The proposed algorithm is a simple alternative solution which can be developed by GIS users with ESRI programming tools (framework). As indicated in the paper, other algorithms generate skeletons of entire polygons, which are then modified in successive steps of the process, according to needs. The proposed algorithm introduces a reverse approach: a polygon skeleton with the expected shape is generated based on a set of base points selected by the user. In our opinion, the strength of the developed solution lies in its simplicity. The authors were impressed by the Splitarea algorithm which inspired the present study. Based on our previous experiences [14], attempts were made to develop a similar algorithm with input conditions supporting the assumption that the expected shape of a skeleton could be defined already at the stage of data preparation.

 Point 4:   The paper seems to include three research tasks, but the two of these tasks are never formulated in clear and explicit manner. First, the authors (on page 2) speak about assumption that the designed algorithm can be used to generate an appropriately shaped skeleton already at the beginning of the process. The second task focuses on polygons with numerous holes (islands on water bodies), where the skeleton’s location between islands was modified automatically based on the preset criteria. So, the first statement is about the assumption, and the second statement is about the focus. Then, what are the tasks related to these statements? The following content of the paper does not bring a specific explanations on this, and the reader must infer them from description of the methodology and results.

Response 4: Three research tasks were formulated, and the first and most important task was described in greater detail in the revised manuscript.

Point 5:    The main subject of the paper is so-called BSplit algorithm for generating the skeleton. Surprisingly, the algorithm is never presented as a sequence of actions, which allows it to be reproduced. The Table 2 contains some comparison between Splitarea dand BPSplit, but the most complicated stage — skeleton generation is not explained. The paragraph 2.1.3 "Description of the BPSplit algorithm" discusses what the algorithm generates (a skeleton), and on what it is relied (classical skeleton generation methods) or based (rules applied to identified types of polygon segments). Then, what is the full algorithm itself? You should provide the reader with sequence of steps: 1, 2, ... and so on — starting with the source data and ending with generated skeleton.

Response 5: Thank you for this valuable remark. A flowchart with mathematical formulas illustrating the process implemented by the algorithm was included in the revised manuscript. The algorithm was based on framework data. In the revised manuscript, the sequence of operations was presented graphically to complement the mathematical formulas and to improve the clarity of presentation.

Point 6:    The degree of automation of the method is questionable. It seems that selection of the strategy to skeletonize polygon with many holes (ID of the nearest segment vs. similar ID for all island boundaries) relies on user decision. The paper does not discuss the advantages and shortcomings of both approaches, as well as some rules to select between them using automated decision strategy.

Response 6: The proposed approach to modifying a skeleton with numerous islands is simple. In successive modifications, a skeleton can be modified based on island attributes. At present, the strategy for generating a skeleton of a polygon with many holes is selected by the user. The automated decision strategy could definitely be applied for this purpose. Thank you for this wonderful idea! We will consider this strategy in our future attempts to develop the algorithm.

Point 7:    I feel that the smoothing task does not deserve so much attention in the paper. Despite the claim that some alternative smoothing method has been developed, the described strategy is simply the variant of widely used moving average approach.

Response 7: Thank you for this valuable remark. This statement was modified by emphasizing that the alternative smoothing method is a variant of the moving average approach.

Point 8:    The overall style of narrative is hard to consume, because of some flaws in logical sequence of explanations and unusual terminology. For example, the assumptions are presented instead of the tasks themselves. And the term 'apex' is used for what is usually called 'node' or 'vertex' in GIScience.

Response 8: We apologize for these inconsistencies. The relevant corrections were made in the revised manuscript.

 Point 9:   The final section of the paper called Discussion is stylistically more like a Conclusion. However, the Conclusion should not only summarize the results, but also bring some light on what particular advancement your research has brought to scientific community, to highlight the novelty of your method. None of this is present there. And to formulate your novelty, you should first better explain the real problem to solve in Introduction (see Issues 1 and 2).

Response 9: The Discussion and Conclusion sections were revised, and references were made to the Introduction, the authors’ intentions, the research objective, and the results.

The authors would like to thank all Reviewers, in particular Reviewer 3, for the time and effort taken to review this paper. Selected fragments were thoroughly restructured based on the Reviewer’s suggestions, thus improving the clarity and readability of the entire manuscript. 

Round 2

Reviewer 1 Report

Line 21: still, what is a "simple algorithm"??? You did not address this issue in your previous review. Avoid vague adjectives like this - especially in the Abstract! Details on what the algorithm is "simple" about, must be added.

Please check figure numbering, it's all messed up!

Line 164: Figure 5?

Line 169: Figure 6?

etc...

Line 221: Again, check reference format -> replace (Gold 2017) by [6] (if this is the correct reference number).

Line 570 (Figure 26 caption): "... generated in a geometrically complex polygon representing...".

Etc., etc.... well, I shall not repeat my previous review report again in here. Please refer to the PDF I attached in my Round 1 report and check all points carefully.

Author Response

Responses to Reviewers 1 Comments

  1. Please, refer to the attached PDF.

Thank you for the encouraging review and the detailed suggestions in the enclosed file. We did not notice the attached file in the first round of the review process, for which we apologize. The Reviewer’s suggestions were fully considered in the second round, and the legends in all tables were modified accordingly. We would like to thank the Reviewer for the time and effort taken to review the revised manuscript.

Thank you for detailed comments and the time taken to review the revised manuscript.

Reviewer 2 Report

The article "Base Point Split Algorithm for Generating Polygon Skeleton Lines on the Example of Lakes" is still very hard to read:

(1) What is the desired result of the algorithm? What is a good or a bad result? And how to compare them?
(2) The Figures are mixed: Figure #1 is on page 2, followed by Figure #5 on page 4 and Figure #6 on page 5. The next figure is Figure #2 -- distributed over pages 6 and 7.
(3) The only table in the article is "Table 2".

(4) The algorithm outline in Equations (1)-(12) is confusing.

In summary, the topic is interesting and the new algorithm may be a good contribution, but with this revision the article has become worse: it is a mess. Taking a look, for example, at the pages 17 or 18, I ask myself, whether the PDF file is corrupted (by the way, I checked the file using a different system with a different viewer getting the same result).

Author Response

Responses to Reviewers 2 Comments

Thank you for detailed comments and the time taken to review the revised manuscript.

  1. The article "Base Point Split Algorithm for Generating Polygon Skeleton Lines on the Example of Lakes" is still very hard to read:

The algorithm was presented graphically to improve readability. Figure legends were modified according to the Reviewer’s suggestions.

  1. What is the desired result of the algorithm? What is a good or a bad result? And how to compare them?

Various algorithms can be used to generate polygon skeleton lines. There are no good or bad algorithms, only different solutions. However, the multiplicity of the existing algorithms suggests that there is no single ideal solution. The BPSplit algorithm, proposed in our study, was compared with other solutions, in particular the Splitarea algorithm. Four solutions were compared in Figure 7. There are no definitively good or bad solutions, and their applicability should be evaluated by the users. The users can select skeletons with e.g. the smallest length, skeletons that best fit selected polygon boundary lines, or skeletons that best fit  the river line (this approach requires additional hydrographic data). The results generated by different algorithms were not compared. Only the sequence of data processing operations in Splitarea and BPSplit area algorithms were compared.

  1. The Figures are mixed: Figure #1 is on page 2, followed by Figure #5 on page 4 and Figure #6 on page 5. The next figure is Figure #2 -- distributed over pages 6 and 7. The only table in the article is "Table 2".

We apologize for these mistakes. The relevant corrections were made.

  1. The algorithm outline in Equations (1)-(12) is confusing.

The algorithm was presented graphically to improve readability.

  1. In summary, the topic is interesting and the new algorithm may be a good contribution, but with this revision the article has become worse: it is a mess. Taking a look, for example, at the pages 17 or 18, I ask myself, whether the PDF file is corrupted (by the way, I checked the file using a different system with a different viewer getting the same result).

We apologize for the editing errors.  Figure numbers were modified, and formatting errors were corrected.

Reviewer 3 Report

The authors have revised the paper and addressed the issues, which I raised in the first review. The quality of presentation improved, however I still have several notes, which I indicate below.

GENERAL / METHODOLOGICAL ISSUES:

  1. In a revised version of the paper the notion of a "complex" polygon is used. How do you define that polygon is complex? What criteria is used to evaluate its complexity? According to Simple Features OGC/ISO standard even the polygon with numerous holes is just a polygon.

  2. I am still unsatisfied with formulations of the tasks. On page 2 we read that "The first task relied on the assumption that the designed algorithm can be used to generate an appropriately shaped skeleton already at the beginning of the process, and that it can be applied in various cases". This is a statement of the assumption, not the task itself. The correct form would be: "The first task is to investigate whether the designed algorithm can generate appropriately shaped skeleton already at the beginning of the process, and that it can be applied in various cases". The same holds for the task 2. On page 2 we read: "The second research task focused on polygons with numerous holes (islands on water bodies), where the skeleton’s location between islands was modified automatically based on the preset criteria." Still there is no task here, but the focus instead. The correct task formulation would be: "The second research task is to investigate the ability of the proposed algorithm to generate skeltons for polygons with numerous holes (islands on water bodies), where the skeleton’s location between islands was modified automatically based on the preset criteria."

  3. The algorithm representation on pages 3-4 is hard to interpret and in fact is superfluos, since the output of each line is duplicated as input on one of the following lines. I highly recommend to draw this algorithm as a standard algorithm flowchart (i.e. representing it in graphical form).

  4. Figures 5 and 6 contain screenshots of ArcMap tabular interface. I recommend redrawing them as high-quality tables. This will look more professional ans allow you to increase the readability of information represented inside.

  5. As a suggestion to improve the level of automation for your methodology: once the skeleton in formed, you can just caculate pairwise routes between all base points using skeleton as network datase. Then select skeleton edges that participate in these routes. That will remove the necessity to decide between different skeleton generation strategies.

SPECIFIC ISSUES (page / row number):

(1 / 13-14) What are "complex skeleton polygons" mentioned there? Do you mean "skeletons of complex polygons"?

(3 / 92) "algorithms developed by the authors in Python" — replace with "algorithms implemented by the authors in Python". Development of the algorithm is a theoretical task, while Python is used for its implementation.

(20 / 575) Replace "Meisner" with "Meijers"

(20 / 579) "points will also be the hanging edges" — one point (node) cannot be the edge itself. Each edge always consists of two nodes.

Author Response

Responses to Reviewers 3 Comments

Thank you for detailed comments and the time taken to review the revised manuscript.

  1. The authors have revised the paper and addressed the issues, which I raised in the first review. The quality of presentation improved, however I still have several notes, which I indicate below.

Thank you for these encouraging remarks.

GENERAL / METHODOLOGICAL ISSUES:

  1. In a revised version of the paper the notion of a "complex" polygon is used. How do you define that polygon is complex? What criteria is used to evaluate its complexity? According to Simple Features OGC/ISO standard even the polygon with numerous holes is just a polygon.

The concept of a “complex” polygon was introduced based on another reviewer’s suggestion to highlight the presence of universal solutions. The study analyzed several polygons (four lakes), polygons with holes, and polygons with touching holes.

  1. I am still unsatisfied with formulations of the tasks. On page 2 we read that "The first task relied on the assumption that the designed algorithm can be used to generate an appropriately shaped skeleton already at the beginning of the process, and that it can be applied in various cases". This is a statement of the assumption, not the task itself. The correct form would be: "The first task is to investigate whether the designed algorithm can generate appropriately shaped skeleton already at the beginning of the process, and that it can be applied in various cases". The same holds for the task 2. On page 2 we read: "The second research task focused on polygons with numerous holes (islands on water bodies), where the skeleton’s location between islands was modified automatically based on the preset criteria." Still there is no task here, but the focus instead. The correct task formulation would be: "The second research task is to investigate the ability of the proposed algorithm to generate skeltons for polygons with numerous holes (islands on water bodies), where the skeleton’s location between islands was modified automatically based on the preset criteria."

Thank you for these valuable suggestions. All of the Reviewer’s remarks regarding task formulation were considered in the second round of the review process.

  1. The algorithm representation on pages 3-4 is hard to interpret and in fact is superfluos, since the output of each line is duplicated as input on one of the following lines. I highly recommend to draw this algorithm as a standard algorithm flowchart (i.e. representing it in graphical form).

The algorithm was presented graphically to improve readability.

  1. Figures 5 and 6 contain screenshots of ArcMap tabular interface. I recommend redrawing them as high-quality tables. This will look more professional ans allow you to increase the readability of information represented inside.

Figures 5 and 6 were redrawn, the tables were modified and created separately in Excel.

  1. As a suggestion to improve the level of automation for your methodology: once the skeleton in formed, you can just caculate pairwise routes between all base points using skeleton as network datase. Then select skeleton edges that participate in these routes. That will remove the necessity to decide between different skeleton generation strategies.

This is a very interesting suggestion. The use of network analysis tools in the process of simplifying the skeleton was described in Figure 6c. We agree that this approach could be used for generating skeletons for various purposes, to automatically remove loops from the skeleton.

SPECIFIC ISSUES (page / row number):

  1. (1 / 13-14) What are "complex skeleton polygons" mentioned there? Do you mean "skeletons of complex polygons"?

We apologize for this mistake. The relevant corrections were made in the entire manuscript.

  1. (3 / 92) "algorithms developed by the authors in Python" — replace with "algorithms implemented by the authors in Python". Development of the algorithm is a theoretical task, while Python is used for its implementation.

Thank you for this valuable observation. The sentence was modified accordingly.

  1. (20 / 575) Replace "Meisner" with "Meijers"

We apologize for this error.

  1. (20 / 579) "points will also be the hanging edges" — one point (node) cannot be the edge itself. Each edge always consists of two nodes.

We completely agree, and we apologize for this error. Base points are the end nodes of the skeleton’s hanging edges. The relevant correction was made in the revised manuscript.

This manuscript is a resubmission of an earlier submission. The following is a list of the peer review reports and author responses from that submission.